# Psychopathic tendencies are selectively associated with reduced emotional awareness in the context of early adversity

**Ryan Smith** [1]*, **Anne E. Chuning**[1], **Colin A. Tidwell**[2], **John J. B. Allen**[2], **Richard D. Lane**[2,3]

**1** Laureate Institute for Brain Research, Tulsa, Oklahoma, United States of America, **2** Department of Psychology, University of Arizona, Tucson, Arizona, United States of America, **3** Department of Psychiatry, University of Arizona, Tucson, Arizona, United States of America

* rsmith@laureateinstitute.org

**Data Availability Statement:** All relevant data are within the manuscript and its Supporting Information files.

## Abstract

It is unclear at present whether psychopathic tendencies are associated with lower or higher levels of emotional awareness (EA). Given that psychopathy includes a proficiency for manipulating others, one might expect an elevated ability to identify and use information about others' emotions. On the other hand, empathic deficits in psychopathy could arise from reduced emotional awareness. Further, heterogeneity in psychopathy may also play a role, wherein 'secondary' psychopathy is associated with early adversity and high negative affect, while 'primary' psychopathy is not. In this paper, we tested the relationship between EA and psychopathic tendencies in 177 undergraduate students (40 males) who completed the levels of emotional awareness scale (LEAS), the triarchic psychopathy measure (TPM), the affective (empathy-related) subscales of the interpersonal reactivity index (IRI), and two measures of early adversity: the childhood experiences of care and abuse questionnaire (CECA) and the childhood trauma questionnaire (CTQ). We found that lower LEAS scores were associated with higher TPM and lower IRI empathy scores, but these relationships were primarily present in those with early adversity and high negative affect. This suggests that lower EA may be selectively associated with higher levels of secondary psychopathy, while those with higher levels of primary psychopathy remain capable of higher EA.

## Introduction

Psychopathy refers to a constellation of antisocial personality traits, such as cruelty, manipulativeness, grandiosity, callousness, and lack of empathy [1]. Individuals with psychopathy are also characterized as being high in egocentrism/narcissism, exploitativeness, impulsivity, aggression, and criminality, while showing low remorse and fear, and shallow affect more generally [2, 3]. Due in part to its link to harmful behavior and criminality, psychopathy has garnered considerable research attention [4, 5]. This research has characterized the maladaptive nature of psychopathy for both the individual and for society, but its potential adaptiveness at the individual level has also been considered [6–10]. For example, some work suggests that psychopathic traits may represent an adaptive short-term sexual strategy [8]; other work has also shown positive associations with attainment of leadership positions [11].

**Funding:** R.S. is supported by the William K. Warren Foundation (https://www. williamkwarrenfoundation.org/) and the National Institute of General Medical Sciences (P20GM121312; https://www.nigms.nih.gov/). The funders had no role in study design, data collection and analysis, decision to publish, or preparation of the manuscript.ata collection and analysis, decision to publish, or preparation of the manuscript.

**Competing interests:** Richard Lane has disclosed an outside interest in the Electronic Levels of Emotional Awareness Scale owned by Equanimity Health Technologies, LLC to the University of Arizona. Conflicts of interest resulting from this interest are being managed by The University of Arizona in accordance with its policies. All other authors have declared that no competing interests exist. There are no patents, products in development or marketed products associated with this research to declare. This does not alter our adherence to PLOS ONE policies on sharing data and materials.

One common question pertains to the relationship between psychopathy and socio-emotional abilities. On the one hand, empathy deficits in psychopathy (i.e., lack of concern for others) might be expected to hinder such abilities. On the other hand, the manipulation skills, superficial charm, and successful attainment of leadership roles in psychopathy each suggest a proficiency in detecting and capitalizing on the emotions of others [7, 9, 11, 12]. Individuals with psychopathic traits have also been purported to use empathy as a manipulative strategy [13–15]. However, the literature on this topic is mixed. For example, while psychopathy is associated with a reduced tendency to feel negative emotion in response to the suffering of others (i.e., affective empathy; sometimes also called affective theory of mind), some studies in those with psychopathic traits have observed intact abilities to recognize the thoughts or feelings of others (i.e., cognitive theory of mind or cognitive empathy; e.g., see [16–18]). Yet, other studies find that psychopathic traits are negatively correlated with these cognitive abilities [13, 14, 19–22]. Studies examining emotional intelligence (EI) also find mixed results, with intact EI present in some studies (and/or assessment measures) but not others (e.g., [23–25]). Many studies of emotion recognition have also observed reduced performance in psychopathic populations for specific emotions [26]. Psychopaths further show a negative perceptual bias towards others–promoting the perception that everyone else is vulnerable and weak [27]. Psychopathic traits are also positively related to alexithymia (i.e., self-reported difficulties in attending to and understanding one's own emotions and associated bodily sensations) and its associated genetic underpinnings [28].

Another complicating factor is the possibility of heterogeneity or distinct subgroups within psychopathic populations. One well-established distinction is between primary and secondary psychopathy–the former being associated with insufficient arousal to emotional cues but no specific relationship to early adversity or negative affect, while the latter is associated with hyperarousal to emotional cues and elevated negative affect, and thought to develop in response to early abuse and neglect [29–39]. Primary psychopathy is characterized by callous, selfish, deceitful, ruthless, and manipulative behaviors (i.e., associated with "successful" psychopaths), while secondary psychopathy is characterized by risk-taking behavior, impulsivity, short-term thinking, antisocial behavior, and violent criminal behavior (and are more frequently found in prison populations than individuals with primary psychopathy). There is growing evidence that these antisocial tendencies have a significant heritable component [40], and that genetic contributions may be greater in individuals with primary psychopathy. For example, epigenetic modifications to oxytocin receptor systems associated with prosocial behaviors have been observed specifically in children with primary psychopathy profiles [41], while longitudinal studies suggest that learning maladaptive emotional behaviors in adverse childhood environments may play an important role in the development of secondary psychopathy [42, 43]. Here it is important to highlight, however, that other non-causal interpretations remain possible (e.g., if psychopathic traits inherited from parents simultaneously promote childhood maltreatment), and the nature of these relationships could also differ in those with distinct psychopathy profiles.

This distinction may be important in relation to emotional abilities. For example, maltreated children appear to show greater responses to emotional faces (consistent with secondary psychopathy), but are less accurate at labelling them in comparison to peers without reported maltreatment [44]. Another study found that men with primary (but not secondary) psychopathy showed elevated lie detection abilities, while the opposite relationship was present in women [45]. Yet, other studies have found emotion recognition deficits in primary psychopathy but not in secondary psychopathy [15, 46]. Thus, the relationship between socio-emotional skills and psychopathy is not completely understood and may be complicated by heterogeneity in both sex and the etiology of psychopathic traits (among other potential factors).

One specific aspect of emotion processing that has not yet been studied in the context of psychopathy pertains to levels of emotional awareness (EA). Trait EA is widely recognized as an important individual difference variable in mental health research [47–49]. Individuals with high levels of EA report granular emotional experiences in themselves and recognize specific emotions in others. This is thought to signify differences in attention to affective (e.g., bodily) sensations and motivations, and differences in the specificity of the emotion concepts an individual has learned during development. In studies using one leading measure, the Levels of Emotional Awareness Scale (LEAS; [50]), high levels of EA have been linked to emotion recognition ability and openness to experience [50–57], while low levels of EA have been observed in multiple affective disorders [58–62]. As with the other emotion-related abilities discussed above, it is unclear if and how psychopathy should be expected to relate to EA. Empathic deficits in psychopathy might support a hypothesis of reduced awareness for the emotions of others. In contrast, the selective focus on self-interest could promote high EA for the self, and prowess in manipulative strategies could suggest intact EA for others. For example, one study found that those with higher EA were more effective liars [63]. Given the findings reviewed above, it also appears plausible that the relationship between EA and psychopathic traits could differ in primary vs. secondary psychopathy. For example, early adversity is negatively related to EA [64], and EA is thought to reflect adaptive early learning [47, 48], both of which could suggest a negative relationship between EA and secondary psychopathy; that is, early adversity could both hinder development of EA and contribute to secondary psychopathy. However, EA is also closely related to theory of mind [65] and has been associated with greater emotion recognition ability [57], whereas the direction of the relationship between primary/secondary psychopathy and these abilities is somewhat inconsistent across studies. Others have also hypothesized that primary and secondary psychopathy differ in their relationships with specific socioemotional skills, such that those with primary psychopathy are suggested to be more skilled at emotional manipulation, emotional control, and spontaneous mentalization, as well as to possess higher emotional intelligence [19, 22, 66, 67].

In this study we addressed the potential relationship between EA and psychopathy and hypothesized that an association would be present. However, based on the work reviewed above, we expected that this relationship could differ between those with vs. without early adversity (i.e., associated with secondary vs. primary psychopathy profiles, respectively; e.g., see [46]). Based on the common link to early learning, we hypothesized that EA would be negatively correlated with psychopathy scores in those with early adversity (i.e., consistent with secondary psychopathy). Given the mixed literature regarding emotional abilities in primary psychopathy, we predicted the presence of a relationship in those with high psychopathy scores without early adversity, but did not have a strong directional hypothesis. We reasoned that high psychopathy in the absence of early adversity might be associated with higher EA in relation to manipulative social abilities, but that it could also be linked to lower EA in relation to the previous findings reviewed above on emotion recognition and theory of mind deficits.

## Methods

### Participants

A convenience sample of 177 students (40 male) at the University of Arizona (mean age = 19.07, SD = 1.82 years), was recruited from Tucson, AZ. Participants gave informed consent and received course credit for their participation. This study was approved by the University of Arizona Institutional Review Board (protocol # 1811122058). As no prior studies evaluated EA and psychopathy, we were unable to anticipate expected effect sizes on which to conduct a power analysis. Therefore, our study initially aimed to recruit 200 participants as a

reasonable pilot sample size. All participants without complete data were excluded before conducting analyses, leading to the final sample size stated above. This final sample of 177 participants would afford 80% power to detect a small-to-moderate effect size ($\rho$ = .21) relationship between EA and psychopathy, given a significance threshold of $p < .05$.

## Measures

The primary variables of interest were the LEAS [50, 51], the Triarchic Psychopathy Measure (TPM; [68]), and two measures of early childhood adversity: the Childhood Experiences of Care and Abuse questionnaire (CECA; [69]) and the Childhood Trauma Questionnaire (CTQ; [70]). As a second measure of the low empathy associated with psychopathy, we also collected data from affective empathy-related scales in the Interpersonal Reactivity Index (IRI; [71]). Additionally, the Positive and Negative Affect Schedule (PANAS-20; [72]) was included to measure expected differences in negative affect associated with primary vs. secondary psychopathy.

The 10-item LEAS [50, 51] is a widely used performance measure of EA. It can be understood to measure the cognitive granularity/complexity with which individuals conceptualize/ understand the emotions of self and others, which is often considered an important aspect of emotional intelligence. This measure presents participants with 10 hypothetical scenarios and asks them to describe the feelings they believe they and another individual would feel. Scoring is based on the types of words chosen to describe feelings. For each scenario: a score of 0 is given to non-emotional words (e.g. confused); a score of 1 is given to words associated with bodily sensations (e.g., "tired"); a score of 2 is given to words describing emotional actions (e.g., "punching") or single valence distinctions (e.g., good/bad) that entail approach/avoidance; a score of 3 is given to single emotion concept words (e.g., "happy," "sad"); and a score of 4 is given when at least 2 specific emotions (e.g., "sad and angry") are described for a single scenario. Self- and other-related responses are scored separately (i.e., from 0–4). A "total" score is then given for each scenario, indexing the higher of the self- and other-related scores, unless a score of 4 is given for both. In that case, a total score of 5 is given for the scenario (as long as the self- and other-related responses are differentiable; for more details see [50]). Other-related scores can be understood to relate to aspects of cognitive empathy or the overlapping construct of theory of mind for emotions. Crucially, while high scores for this subscale reflect greater granularity in how an individual conceptualizes the emotions of others, it does not necessarily indicate greater affective empathy (e.g., a person could be aware that another individual is in distress without experiencing negative emotions in response). The Cronbach's alpha of the LEAS total score is .88 [56]. These present data were scored based on a previously validated computer scoring method [51]. Additionally, all written responses on the LEAS were manually checked to ensure attentive completion of each item (e.g., the content of written responses directly answered the prompt provided, absence of non-identical responses across items, lack of jumbled letters or non-words, etc.).

The TPM was selected to measure psychopathy within this study because it is based on the contemporary triarchic model of psychopathy, an effort to integrate differing concepts of psychopathy proposed by Patrick and colleagues [73]. Other self-report measures of psychopathy, such as the Psychopathic Personality Inventory (PPI; [74]), the Levenson Self-Report Psychopathy Scale (LSRP; [34]), and the Antisocial Process Screening Device (APSD; [75]), vary in their conceptualization of psychopathy and its traits, leading to potential inconsistencies when comparing results across different studies. While we appreciate these tools, their specific origins, and this existing body of work, we chose the TPM here because it offers a balanced measure of multiple distinct dimensions of psychopathy. For a more in-depth discussion and

review of the differences between leading self-report psychopathy measures, the interested reader may see the following: [76–79]. Additionally, in line with our hypothesis about differences in primary and secondary psychopathy, the TPM includes two subscales related to these two presentations. Higher boldness is typically associated with primary psychopathy, while higher impulsiveness is typically associated with secondary psychopathy [80]. However, we were also interested in the relationship between EA and overall levels of psychopathic tendencies. Therefore, initial analyses focused on total scores; further analyses then examined subscale scores to better understand the specific dimensions of psychopathy that best explained initially observed patterns of relationships between variables.

The TPM [68] is a 58-item self-report measure that includes three subscales: boldness (19 items), meanness (19 items), and disinhibition (20 items). Each item is on a 4-point Likert scale and asks participants to rate the degree to which each item applies to them ('mostly false', 'false', 'mostly true' and 'true'). Meanness corresponds to tendencies toward callousness, cruelty, predatory aggression, and excitement-seeking. Disinhibition corresponds to tendencies toward impulsiveness, irresponsibility, oppositionality, and anger/hostility. Boldness corresponds to patterns of interpersonal behavior (dominance, persuasiveness, social assurance), emotional dispositions (self-assurance, resiliency, optimism), and how venturesome an individual is (intrepidness, courage, tolerance for uncertainty). Cronbach's alpha values are .80 for both the Boldness and Disinhibition subscales, and .87 for the Meanness subscale.

The CECA and the CTQ were selected to measure early adversity, as both follow the conceptual framework of the initial Adverse Childhood Experiences (ACES) study [81]. The CTQ has good convergent validity with the ACES measure and research has shown that it is more sensitive at capturing different types of early adversity [82]. The CECA [69] measures poor parental care with 'neglect' and 'antipathy' subscales. Each subscale has eight items and is scored separately for mother and father figures (32 items total). As one example item, 'she was very difficult to please' would be rated on a five-point scale (1 = 'no, not at all', 3 = 'unsure' and 5 = 'yes, definitely'). We also took the mean of the subscale scores to generate a total CECA score. The CECA has good validity and reliability, with Cronbach's alpha values of .80 for the antipathy subscales and .81 for the neglect subscales [69]. We used subscales of the CTQ [70] measuring physical, emotional, and sexual abuse, each consisting of five items scored on a five-point scale (1 = 'never true' and 5 = 'very often true'). The mean of the subscale scores generated a total CTQ score. Cronbach's alpha values for the CTQ subscales are .83 for physical abuse, .87 for emotional abuse, and .93 for sexual abuse [70].

The PANAS-20 [72] is a widely used measure of positive and negative affective states. It consists of 20 words that describe different feelings and emotions (10 negative and 10 positive valence; e.g., "Guilty", "Excited"). For each word, participants are asked to indicate, on a scale ranging from 1 ("very slightly to not at all") to 5 ("extremely"), the "extent to which you generally feel this way, that is, how you feel on the average". Items are summed for each valence to give a total score for positive and negative affect, respectively. Aside from early adversity measures, this was used as a way to further differentiate individuals better corresponding to the construct of primary vs. secondary psychopathy (i.e., where only the latter would be expected to have high levels of negative affect; [30–32]). The PANAS-20 demonstrates good validity and reliability, with Cronbach's alpha of .88 for positive affect and .87 for negative affect [72].

The IRI [71] assesses both cognitive and affective components of dispositional empathy (2 subscales each). Based on our a priori hypotheses, and the study's time constraints, we only assessed the two affective subscales: Empathic Concern (having sympathy for others in need) and Personal Distress (having negative arousal in response to perceived distress in others). While the two cognitive subscales had the potential to provide secondary insights, they were not prioritized because they were not required to assess the hypothesized relationship between

EA, early adversity, and psychopathy, as well as the low levels of affective empathy associated with psychopathy more broadly. Each subscale has seven items. An example item for Empathic Concern is: "I often have tender, concerned feelings for people less fortunate than I". An example item for Personal Distress is: "When I see someone who badly needs help in an emergency, I go to pieces". Participants responded on a 5-point scale, ranging from 0 (doesn't describe me well) to 4 (describes me very well). The Empathic Concern subscale acted as a convergent measure of the low affective empathy associated with psychopathy; the content of this subscale overlaps with the Meanness dimension measured in the TPM. The Personal Distress subscale does not pertain to the lack of concern for others characteristic of psychopathy, so it was used primarily for exploratory purposes in assessing dispositions toward negative affect (i.e., complementary to the PANAS). Both subscales demonstrate good reliability, as the Empathic Concern subscale has shown a Cronbach's alpha of .71 and the Personal distress subscale has shown a Cronbach's alpha of .76 [71].

We note here that data on LEAS, CTQ, and CECA scores have been analyzed in a previous study [64]. The data shown from TPM, IRI, and PANAS scores, and their relations to EA and early adversity, are novel to this study.

Patterns of responses on all questionnaires were manually checked to ensure a lack of random or inattentive responding. This led to the removal of partial data from six unique participants who showed identical responses across all items of a given scale. Final Ns for the following measures were: TPM (N = 176), IRI (N = 173), CTQ (N = 172), CECA (N = 173), PANAS (N = 173).

## Analyses

To assess our primary question regarding the overall relationship between EA and psychopathy, we ran JZS Bayes factor analyses with default prior scales in R (BayesFactor package [83, 84]) comparing null (intercept only) models of either TPM or IRI empathic concern scores to the space of regression models that included all combinations of main effects of age, sex, either CTQ or CECA Total scores, LEAS Total scores, and potential pair-wise interactions (models including interactions without main effects were not included). A BF represents the ratio of the probability of observed data under one model vs. another (i.e., where a higher probability of data under a model provides more evidence for that model). For example, BF = 1/3 indicates that data are three times more likely under the null model than the alternative model, BF = 1 indicates equal evidence for the null and alternative models, and BF = 3 indicates that the data are three times more probable under the alternative model than the null model. When interpreting the strength of evidence of each result, we adopt the guidelines described in Lee and Wagenmakers [85]: BF = 1–3, poor/anecdotal evidence for the alternative hypothesis; 3–10, moderate evidence; 10–30, strong evidence; 30–100, very strong evidence; >100, extremely strong evidence. In cases of extremely strong evidence, we will indicate BF > 100 (as these values can get cumbersomely large under default priors in large sample sizes or when one model fits much better than another); we further compare BFs for the 1st- and 2nd-best model to offer a better sense of the relative evidence of the most competitive models. These BF analyses provide several advantages. Most importantly here, they provide a straightforward basis for model selection (i.e., indicating which predictors are relevant to include), and they allow the evaluation of evidence for the null model as well as for alterative models. Thus, they improve interpretability by allowing us to gain confidence that a relationship between variables is absent vs. present (i.e., as opposed to simply failing to reject a null hypothesis).

After these planned analyses, we performed post-hoc correlations and t-tests to provide insights about which subscales for each measure contributed most to our findings for the respective total scores. To further interpret our primary results and address the more specific

question about heterogeneity in EA within high-psychopathy individuals, we also reperformed some analyses focused only on individuals with high TPM scores (based on a median split, described further below). Similar complementary analyses were also done for IRI Empathic Concern scores.

It is important to note here that we do not examine the relationship between LEAS, CTQ, and CECA scores, because this has already been addressed in a previous study using this dataset [64]. Therefore, we focus only on novel analyses of the relationship between these measures and TPM, IRI, and PANAS scores.

## Results

Table 1 shows the descriptive statistics for each measure, separated by sex, as well as p-values, effect sizes (Cohen's d), and Bayes factors for post-hoc t-tests indicating whether scores differed for males and females. This is shown because several models below revealed significant effects of sex. These differences in males and females are also expected based on previous literature [57, 76, 86].

### Emotional awareness, psychopathy, and early adversity

In a Bayesian regression analysis assessing age, sex, CECA Total scores, and LEAS Total scores (and their interactions) as possible predictors of TPM Total scores, the most evidence was

**Table 1. Summary statistics (mean and SD) for study measures by sex.**

| Measures[a] | Females | Males | $p$[b] | Cohen's $d$ | Bayes Factor |
|---|---|---|---|---|---|
| | (N = 137) | (N = 40) | | | |
| TPM Total | 58.07 (16.34) | 71.33 (17.43) | **<0.001** | **0.80** | **>100** |
| TPM Boldness | 28.94 (8.45) | 32.20 (7.69) | **0.03** | **0.39** | **1.65** |
| TPM Meanness | 10.32 (7.49) | 17.55 (8.58) | **<0.001** | **0.93** | **>100** |
| TPM Disinhibition | 16.23 (8.22) | 18.58 (9.71) | 0.13 | 0.27 | 0.57 |
| IRI Empathic Concern | 22.25 (4.15) | 18.58 (4.12) | **<0.001** | **0.89** | **>100** |
| IRI Personal Distress | 13.10 (4.05) | 11.85 (4.39) | 0.096 | 0.30 | 0.68 |
| Negative Affect | 23.35 (7.71) | 22.08 (8.46) | 0.37 | 0.16 | 0.28 |
| Positive Affect | 34.06 (6.60) | 36.08 (6.94) | 0.063 | 0.30 | 0.83 |
| Age | 18.99 (1.71) | 19.38 (2.12) | 0.233 | 0.18 | 0.37 |
| LEAS Total | 33.81 (4.06) | 31.62 (3.28) | **0.002** | **0.47** | **14.80** |
| LEAS Self | 29.58 (4.41) | 26.32 (4.34) | **<0.001** | **0.62** | **>100** |
| LEAS Other | 27.50 (3.97) | 25.42 (4.91) | **0.007** | **0.42** | **5.61** |
| CECA Total | 15.86 (5.09) | 16.89 (4.91) | 0.263 | 0.20 | 0.34 |
| CECA Mother Antipathy | 17.88 (6.11) | 18.92 (5.27) | 0.335 | 0.18 | 0.30 |
| CECA Mother Neglect | 11.65 (5.39) | 13.46 (6.34) | 0.078 | 0.32 | 0.79 |
| CECA Father Antipathy | 18.24 (6.90) | 18.62 (5.15) | 0.753 | 0.06 | 0.22 |
| CECA Father Neglect | 15.66 (7.48) | 16.56 (7.40) | 0.508 | 0.12 | 0.25 |
| CTQ Total | 7.11 (2.65) | 6.88 (2.57) | 0.640 | 0.09 | 0.21 |
| CTQ Physical Abuse | 6.23 (2.61) | 6.67 (2.79) | 0.370 | 0.16 | 0.27 |
| CTQ Emotional Abuse | 9.21 (4.57) | 8.21 (4.05) | 0.218 | 0.23 | 0.35 |
| CTQ Sexual Abuse | 5.87 (2.78) | 5.77 (2.55) | 0.836 | 0.04 | 0.20 |

[a]Data from the TPM and IRI are novel to this study. Data from other measures (bottom table section) have previously been described [64].

[b]p-values are based on two-sample t-tests between males and females.

[c]Note that, after quality control checks, the final Ns for the following measures were: TPM (N = 176), IRI (N = 173), PANAS (N = 173), CTQ (N = 172), CECA (N = 173).

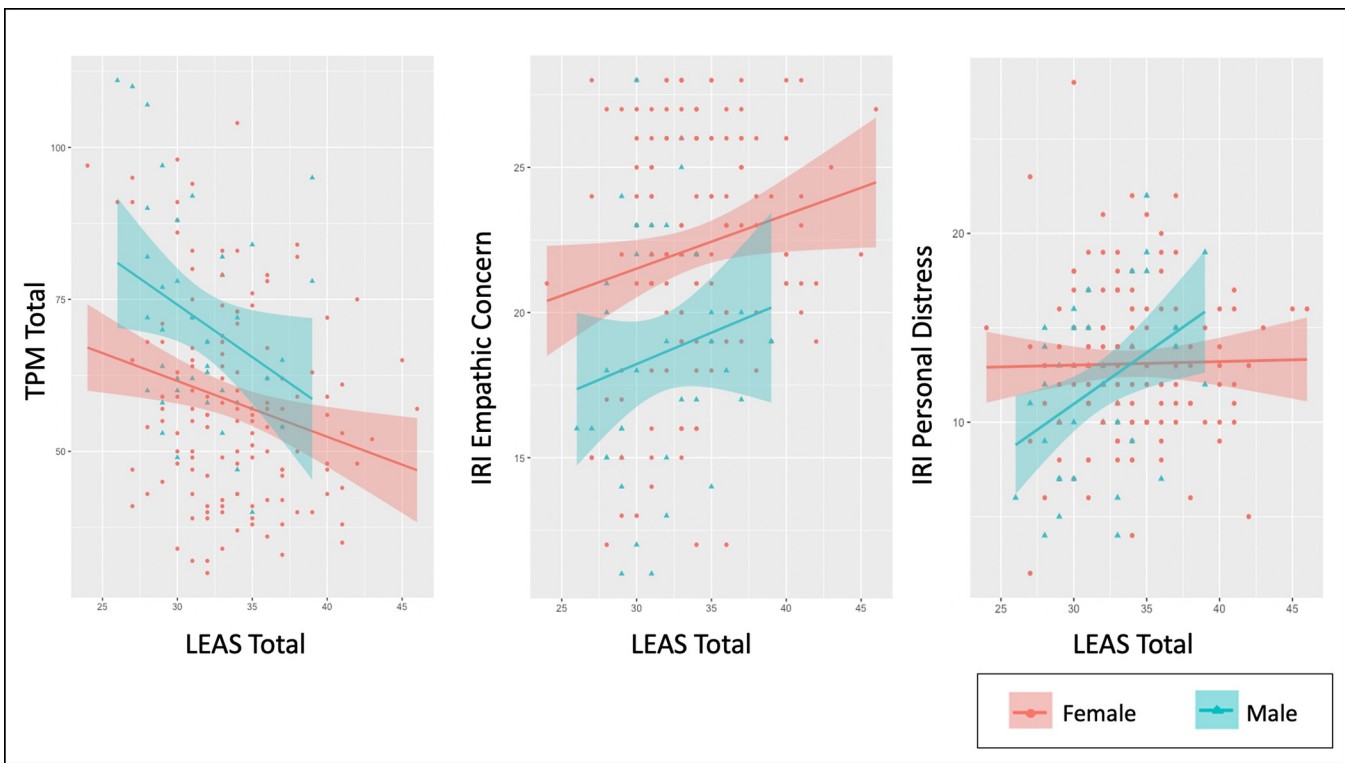

**Fig 1. Relationships between emotional awareness, psychopathy, and empathy measures.** This figure shows correlations between Levels of Emotional Awareness (LEAS) Total scores and both Triarchic Psychopathy Measure (TPM) Total scores and Interpersonal Reactivity Index (IRI) Empathic Concern and Personal Distress scores, separated by sex.

found for a model including sex (b = -4.86, CI = [-7.63, -2.10]), CECA Total scores (b = 0.89, CI = [0.45, 1.34]), and LEAS Total scores (b = -0.81, CI = [-1.39, -0.21]): BF > 100 relative to an intercept-only model (extremely strong evidence). The 2nd-best model added an interaction between CECA Total scores and LEAS Total Scores (BF = 0.70 relative to the winning model). Post-hoc analyses showed a negative relationship between LEAS Total scores and TPM Total scores ($r$ = -.30, $p$ < .001, BF > 100; see **Fig 1**), a positive relationship between CECA Total scores and TPM Total scores ($r$ = .31, $p$ < .001, BF > 100), and greater TPM Total scores in males than females ($t$(174) = -4.36, 95% CI [-17.37, -6.54], $p$ < .001, $d$ = 0.79, BF > 100). The interaction between LEAS and CECA scores in the 2nd-best model was driven by a stronger negative relationship between LEAS scores and TPM scores in those with higher CECA scores. This could be visualized by taking a median split on CECA scores (low < 14.75, high ≥ 14.75) and correlating LEAS and TPM scores for those with high vs. low CECA scores separately (see **Fig 2**). Although displayed here primarily for purposes of visualization, we note that the correlation in those with high CECA scores was $r$ = -.38 ($p$ < .001, BF > 100), while the correlation in those with low CECA scores was $r$ = -.12 ($p$ = .29, BF = .42).

To better understand the relationship between these measures, we subsequently conducted post-hoc correlations between the subscales for each of these measures (shown in **Fig 3**). These results suggested that the negative relationship with LEAS Total scores (and with both Self and Other subscales) was driven by the TPM Meanness and Disinhibition subscales, but not by the Boldness subscale (with evidence favoring the absence of this relationship, BF = .25), and that the positive relationship with CECA Total scores (and all subscales) was also driven by the TPM Meanness and Disinhibition subscales, but not by the Boldness subscale (with evidence

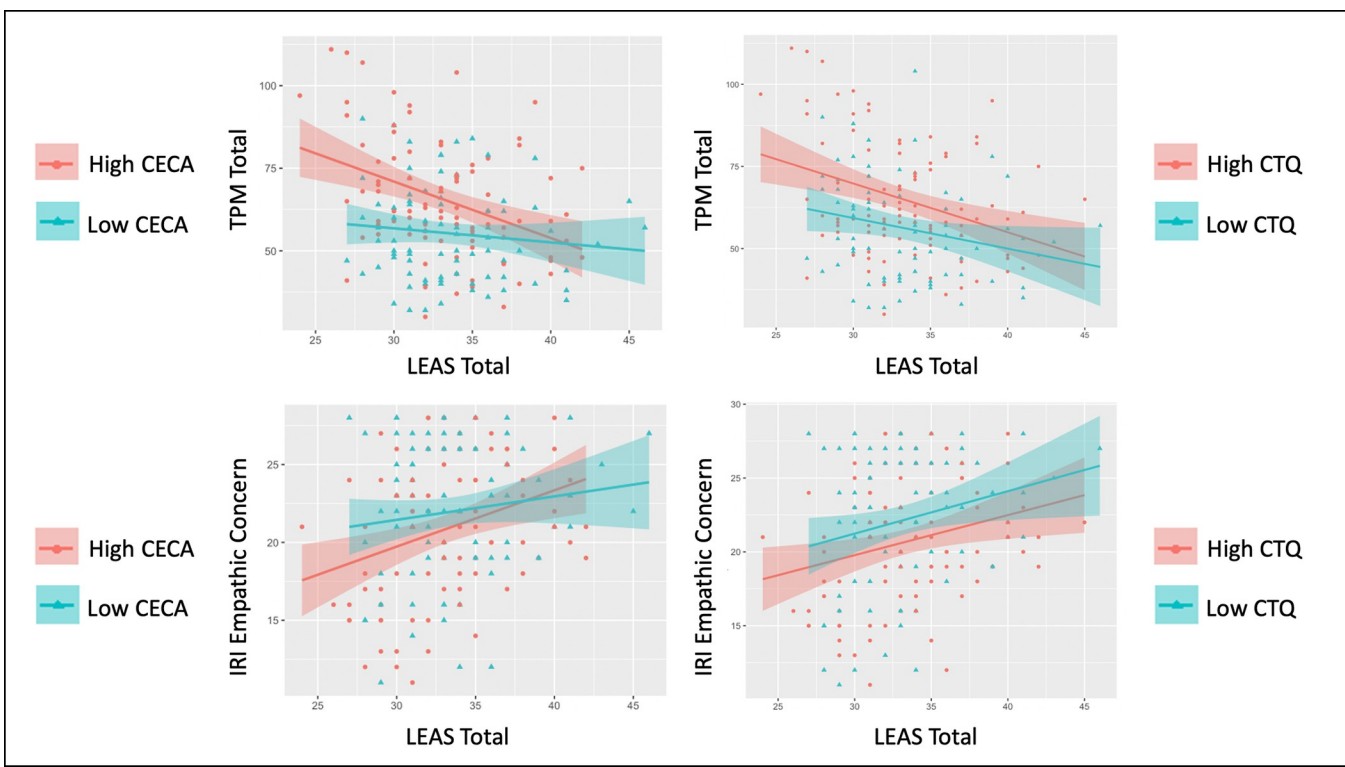

**Fig 2. Effects of early adversity on the relationships between emotional awareness, psychopathy, and empathy measures.** This figure shows correlations between Levels of Emotional Awareness (LEAS) Total Scores and both Triarchic Psychopathy Measure (TPM) Total scores and Interpersonal Reactivity Index (IRI) Empathic Concern scores, separated by high vs. low levels of early adversity (based on median splits). Early adversity measures included the Childhood Experiences of Care and Abuse questionnaire (CECA) and the Childhood Trauma Questionnaire (CTQ). The High-CECA group had 24/40 males and 65/133 females. The High-CTQ group had 21/39 males and 72/133 females. This visual comparison between individuals with high and low levels of early adversity illustrates the significant interaction observed in the main analyses between emotional awareness and early adversity in predicting psychopathy and empathic concern.

favoring the absence of this relationship, BF = .18). Notably, LEAS Total scores were not significantly related to IRI personal distress scores, consistent with prior studies indicating that, unlike alexithymia measures (e.g., [87]), LEAS is independent of negative affect [88]. Similarly, LEAS Total scores were also not related to PANAS negative affect scores ($r$ = -.08, $p$ = .3, BF = .3) or positive affect scores ($r$ = .06, $p$ = .42, BF = .24). IRI Empathic Concern and Personal Distress scores were significantly correlated ($r$ = .16, $p$ = .04, BF = 1.43).

In a Bayesian regression analysis assessing age, sex, CTQ Total scores and LEAS Total scores (and their interactions) as possible predictors of TPM Total scores, the most evidence was found for a model including sex (b = -5.79, CI = [-8.53, -3.08]), CTQ Total scores (b = 1.27, CI = [0.29, 2.25]), LEAS Total scores (b = -0.69, CI = [-1.28, -0.12]), and an interaction between CTQ and LEAS Total scores (b = -0.31, CI = [-0.53, -0.09]): BF > 100 relative to an intercept-only model (extremely strong evidence). The 2nd-best model also included an interaction between sex and LEAS Total scores (BF = 0.48 relative to the winning model). A post-hoc Pearson correlation analysis showed a positive relationship between CTQ Total scores and TPM Total scores ($r$ = .32, $p$ < .001, BF > 100). The interaction between LEAS and CTQ scores was driven by a stronger negative relationship between LEAS scores and TPM scores in those with higher CTQ scores. This could be visualized by taking a median split on CTQ scores (low < 6, high ≥ 6) and correlating LEAS and TPM scores for those with high vs. low CTQ scores separately (**Fig 2**). Although displayed here primarily for purposes of

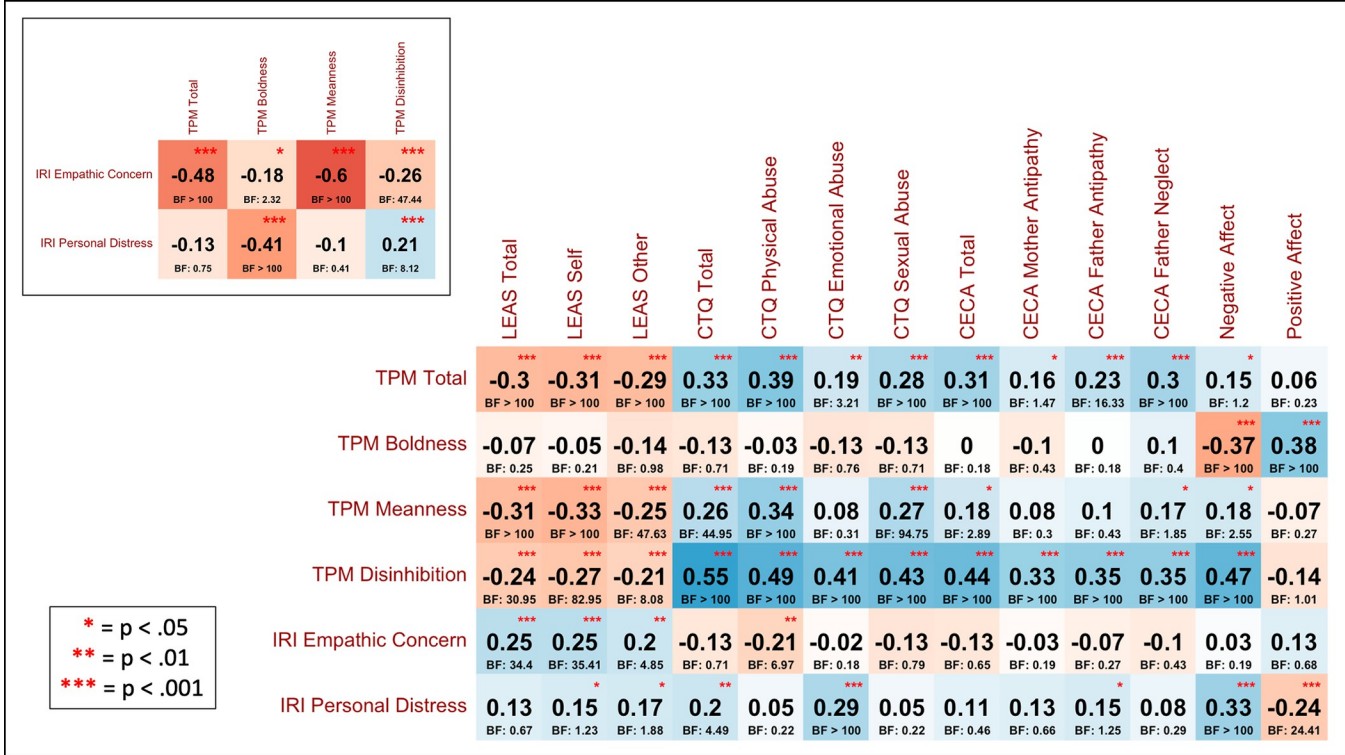

**Fig 3. Correlations between measures.** This figure shows post-hoc correlations (and associated BFs) across the full sample between additional subscales, illustrating their contributions to the main findings described in the text. Measures include the Levels of Emotional Awareness scale (LEAS), Triarchic Psychopathy Measure (TPM), Interpersonal Reactivity Index (IRI), Childhood Experiences of Care and Abuse questionnaire (CECA), Childhood Trauma Questionnaire (CTQ), and the Positive and Negative Affect scales from the Positive and Negative Affect Schedule (PANAS-20).

visualization, we note that the correlation in those with high CTQ scores was $r = -.33$ ($p = .001$, BF = 37.3), while the correlation in those with low CTQ scores was $r = -.23$ ($p < .05$, BF = 1.72). Post-hoc correlations (**Fig 3**) suggested that the positive relationship with CTQ Total scores (and all subscales) was driven by the TPM Meanness and Disinhibition subscales, but not by the Boldness subscale (with evidence favoring the absence of this relationship, BF = .71).

### Emotional awareness, empathy, and early adversity

**IRI empathic concern.** Identical analyses to those reported for TPM scores above were also performed assessing the relationship between LEAS Total scores and IRI Empathic Concern scores. Detailed results are provided in **Supplementary Materials**. Briefly, Bayesian regression analyses found evidence for a similar pattern of results (see **Figs 1 and 2**), including a positive relationship between LEAS Total scores and IRI Empathic Concern scores, a negative relationship between CECA/CTQ Total scores and IRI Empathic Concern scores, greater IRI Empathic Concern scores in females than males, and a stronger positive relationship between LEAS scores and IRI Empathic Concern scores in those with higher CECA/CTQ scores. This could be visualized by taking the same median split on CECA/CTQ scores as in our previous analyses and correlating LEAS and IRI Empathic Concern scores for those with high vs. low CECA/CTQ scores separately (see **Fig 2**; although this interaction was less visually apparent using the median split on CTQ scores). The negative relationship between IRI Empathic Concern and CECA/CTQ Total scores was driven primarily by CECA Mother

Neglect scores and by CTQ physical and sexual abuse scores, but not by emotional abuse scores (**Fig 3**).

**IRI personal distress.** For the interested reader, we provide detailed results of analyses (analogous to those above) for IRI personal distress within **Supplementary Materials**. There was a notable interaction between sex and LEAS Total scores in predicting IRI Personal Distress scores, which showed a positive relationship in males ($r = .41$, $p = .009$, BF = 6.86), but no relationship in females ($r = .02$, $p = .83$, BF = .20; see **Fig 1**). These analyses were considered exploratory, as we did not have strong a priori hypotheses regarding this measure. However, given the observed pattern of relationships between this measure and CTQ emotional abuse, TPM Boldness, and PANAS scores (see **Fig 3**), it might be viewed in this context as a convergent measure of dispositional negative affect (i.e., more consistent with the high-anxiety states associated with secondary psychopathy [89]).

## Secondary analyses in high-psychopathy participants

To more thoroughly interpret the significant interactions between psychopathy, early adversity, and emotional awareness identified in our primary analyses above, we performed additional post-hoc analyses to selectively probe these relationships in participants with high psychopathy scores. To do so, we performed a median split on TPM scores (low < 58.5, high ≥ 58.5) and then restricted analyses to high-TPM participants. We then used the same analysis approach as above to test if the interaction between LEAS scores and early adversity scores was still present in predicting psychopathy scores. This allowed us to test for evidence of heterogeneity in EA within high-psychopathy individuals in relation to early adversity (i.e., consistent with primary vs. secondary psychopathy). Note that our question here was specifically about the possibility of distinct patterns in individuals showing high psychopathy scores; thus, the low-psychopathy individuals (below the median for TPM scores; 10/40 males, 78/136 females) were not included in these analyses.

Results were similar to our main analyses above. Specifically, using the CECA, the most evidence was found for a model including CECA Total scores (b = 4.91, CI = [-.09, 10.01]), LEAS Total scores (b = 1.57, CI = [-1.16, 4.31]), and an interaction between CECA Total scores and LEAS Total scores (b = -.14, CI = [-0.29, 0.01]): BF = 18.20 relative to an intercept-only model (strong evidence). The 2nd-best model removed the interaction (BF = 0.62 relative to the winning model). Using the CTQ, the most evidence was found for a model including CTQ Total scores (b = 11.52, CI = [4.55, 18.54]), LEAS Total scores (b = 1.86, CI = [0.12, 3.60]), and an interaction between CTQ and LEAS Total scores (b = -0.35, CI = [-0.58, -0.12]): BF > 100 relative to an intercept-only model (extremely strong evidence). The 2nd-best model added a main effect of sex and an interaction between sex and CTQ Total scores (BF = 0.89 relative to the winning model).

As above, for further interpretation and visualization of these interactions (see **Fig 4**), the high-psychopathy individuals were then divided into groups with low vs. high levels of early adversity, which allowed additional examination of EA in those with profiles more consistent with primary vs. secondary psychopathy. To do so, we performed a median split on CTQ (low < 6, high ≥ 6) and CECA (low < 14.75, high ≥ 14.75) Total scores. We then divided participants into two high-TPM groups for each measure: high-TPM/high-early adversity (CTQ: 15/39 males, 39/133 females; CECA: 18/40 males, 36/133 females) and high-TPM/low-early adversity (CTQ: 14/39 males, 17/133 females; CECA: 11/40 males, 20/133 females).

Illustrating the significant interactions described above, **Fig 4** shows that those with high psychopathy and high early adversity showed significant negative relationships between TPM Total scores and LEAS Total Scores (high-TPM/high-CECA: $r = -.35$, $p = .02$, BF = 3.30; high-

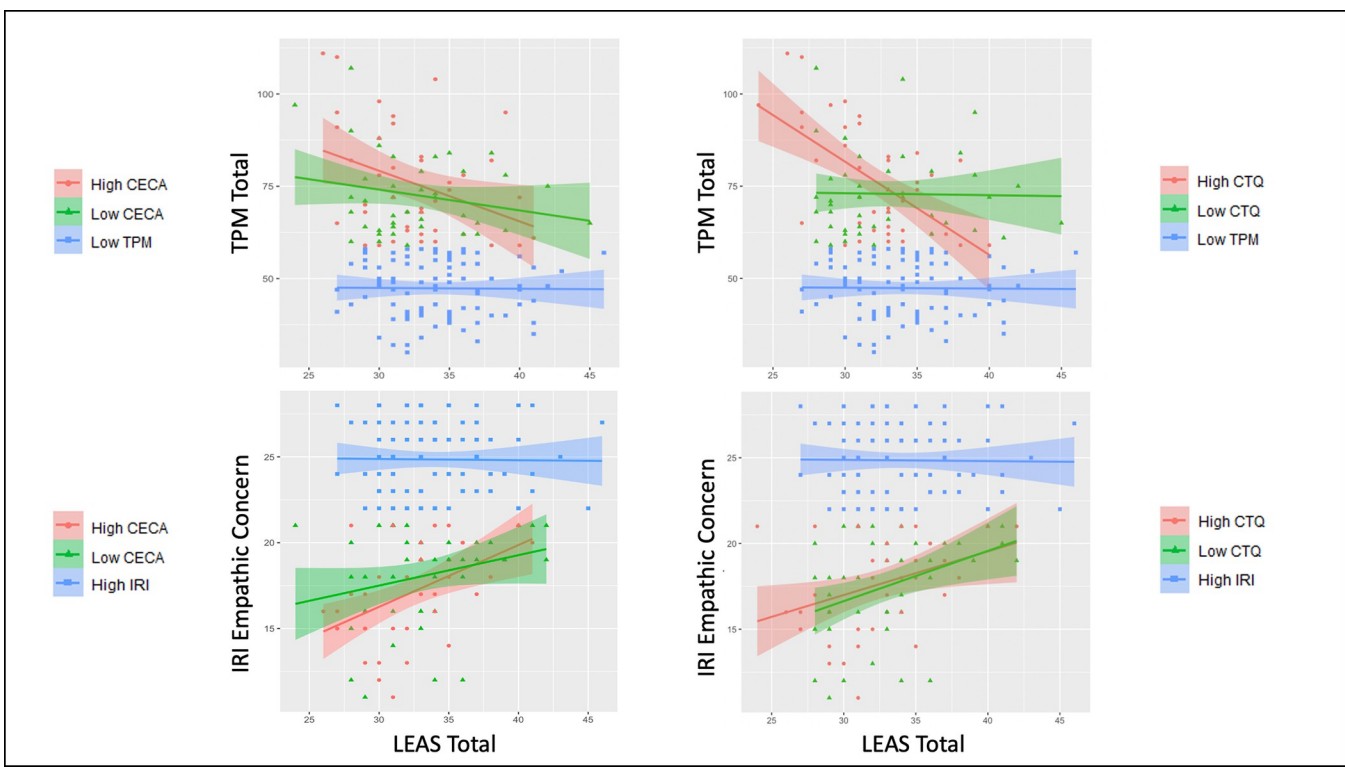

**Fig 4. Analysis of individuals with high levels of psychopathy with vs. without early adversity.** This figure shows correlations in high-psychopathy/low-empathy groups between Levels of Emotional Awareness (LEAS) Total Scores and both Triarchic Psychopathy Measure (TPM) Total scores and Interpersonal Reactivity Index (IRI) Empathic Concern scores, separated by high vs. low levels of early adversity (based on median splits). Early adversity measures included the Childhood Experiences of Care and Abuse questionnaire (CECA) and the Childhood Trauma Questionnaire (CTQ). Unlike in **Fig 2**, high and low early adversity levels were taken only from individuals with scores above and below the median for the TPM and IRI, respectively. Although the focus of these analyses was restricted to high-psychopathy/low-empathy individuals, for the interested reader we also illustrate the absence of any relationships observed in the remaining low-psychopathy and high-empathy individuals (blue lines).

TPM/high-CTQ: $r = -.58$, $p < .001$, BF $> 100$). In contrast, those with high TPM scores but low early adversity showed evidence favoring the absence of a relationship between TPM Total scores and LEAS Total Scores (high-TPM/low-CECA: $r = -.22$, $p = .16$, BF $= .82$; high-TPM/low-CTQ: $r = -.02$, $p = .90$, BF $= .33$). Although not the focus of these analyses, for the interested reader we note that those in the low-psychopathy group also showed no relationship between TPM Total scores and LEAS Total Scores ($r = -.01$, $p = .92$, BF $= .25$). This same pattern of results was also present when analyzing males and females separately (see **S1 Fig in S1 File**). Comparison of high- and low-psychopathy individuals also confirmed that high-psychopathy individuals had experienced greater early adversity overall with respect to CECA Total scores ($t(171) = 3.32$, 95% CI [1.01, 3.96], $p = .001$, $d = 0.51$, BF $= 24.47$) and CTQ Total scores ($t(170) = 2.58$, 95% CI [0.24, 1.80], $p = .011$, $d = 0.39$, BF $= 3.47$).

Consistent with a secondary psychopathy profile, a two-sample t-test further revealed that high-TPM/high-CTQ individuals showed significantly greater negative affect on the PANAS than high-TPM/low-CTQ individuals ($t(82) = 3.11$, 95% CI [2.01, 9.14], $p = .003$, $d = .70$, BF $= 13.70$). In contrast, although statistically significant, an analogous two-sample t-test did not reveal strong evidence for a difference in PANAS negative affect scores in high-TPM/high-CECA vs. high-TPM/low-CECA individuals ($t(82) = 2.17$, 95% CI [0.33, 7.66], $p = .03$, $d = .49$, BF $= 1.74$). That is, while the BF for the CTQ analysis provided strong evidence for the presence of this difference in negative affect, the BF for the CECA analysis only provided poor/

anecdotal evidence in favor of greater negative affect in the high-TPM/high-CECA individuals. Secondary analyses also indicated a similar pattern of results with respect to IRI personal distress scores, suggesting greater dispositions toward distress for high-psychopathy individuals who had experienced early adversity (see **Supplementary Materials**).

A two-sample t-test did not reveal differences in LEAS Total scores between high-TPM/high-CECA individuals and high-TPM/low-CECA individuals ($t(83)$ = -.77, 95% CI [-2.46, 1.09], $p$ = .44, $d$ = .17, BF = .30). Similarly, a two-sample t-test also did not reveal differences in LEAS Total scores between high-TPM/high-CTQ individuals and high-TPM/low-CTQ individuals ($t(83)$ = 1.29, 95% CI [-0.62, 2.90], $p$ = .2, $d$ = .29, BF = .48). These BFs provide positive evidence for the absence of these relationships.

Post-hoc comparisons of other outcome measures between high-TPM/high-early adversity and high-TPM/low-early adversity individuals are shown in **Table 2**. Notably, unlike Meanness and Disinhibition, TPM Boldness was higher in those with Low CECA/CTQ. PANAS Positive Affect was also higher in those with low CECA/CTQ.

**Table 2. Summary statistics (mean and SD) of measures in high-psychopathy group by early adversity level.**

| Measures[a] | High CECA | Low CECA | $p$[b] | Cohen's $d$ | Bayes Factor |
|---|---|---|---|---|---|
| | (N = 54) | (N = 31) | | | |
| TPM Total | 76.07 (14.50) | 70.77 (8.88) | 0.069 | 0.42 | 1.01 |
| TPM Boldness | 32.26 (7.61) | 35.52 (8.43) | 0.07 | 0.41 | 0.99 |
| TPM Meanness | 17.72 (8.39) | 15.81 (6.05) | 0.251 | 0.251 | 0.40 |
| TPM Disinhibition | 23.11 (9.46) | 17.52 (6.64) | **0.005** | **0.66** | **8.27** |
| IRI Empathic Concern | 20.04 (4.45) | 19.10 (3.98) | 0.334 | 0.22 | 0.35 |
| IRI Personal Distress | 12.85 (4.66) | 12.32 (3.49) | 0.587 | 0.12 | 0.27 |
| PANAS Negative Affect | 25.38 (8.67) | 21.39 (7.15) | **0.033** | **0.49** | **1.74** |
| PANAS Positive Affect | 34.02 (6.98) | 35.77 (6.76) | 0.264 | 0.25 | 0.41 |
| Age | 18.96 (1.43) | 18.87 (1.20) | 0.763 | 0.07 | 0.24 |
| LEAS Total | 32.31 (4.07) | 33.00 (3.72) | 0.444 | 0.17 | 0.30 |
| LEAS Self | 27.43 (5.24) | 28.90 (3.96) | 0.177 | 0.30 | 0.52 |
| LEAS Other | 25.69 (4.71) | 26.90 (4.12) | 0.234 | 0.27 | 0.44 |
| Measures | High CTQ | Low CTQ | $p$ | Cohen's $d$ | Bayes Factor |
| | (N = 54) | (N = 31) | | | |
| TPM Total | 75.07 (14.29) | 72.52 (10.17) | 0.383 | 0.20 | 0.33 |
| TPM Boldness | 32.28 (7.92) | 37.23 (6.69) | **<0.001** | **0.79** | **40.90** |
| TPM Meanness | 17.69 (8.29) | 15.87 (6.31) | 0.295 | 0.24 | 0.38 |
| TPM Disinhibition | 23.24 (9.34) | 17.29 (6.72) | **0.003** | **0.70** | **13.69** |
| IRI Empathic Concern | 19.91 (4.10) | 19.32 (4.63) | 0.551 | 0.14 | 0.27 |
| IRI Personal Distress | 13.68 (4.44) | 10.90 (3.29) | **0.003** | **0.69** | **11.12** |
| PANAS Negative Affect | 25.96 (8.19) | 20.39 (7.44) | **0.003** | **0.70** | **13.70** |
| PANAS Positive Affect | 33.92 (6.84) | 35.94 (6.97) | 0.2 | **0.29** | **0.48** |
| Age | 19.00 (1.48) | 18.81 (1.08) | 0.526 | 0.14 | 0.28 |
| LEAS Total | 32.98 (4.30) | 31.84 (3.15) | 0.2 | 0.29 | 0.48 |
| LEAS Self | 28.46 (4.93) | 27.10 (4.64) | 0.213 | 0.28 | 0.46 |
| LEAS Other | 26.56 (5.00) | 25.39 (3.48) | 0.253 | 0.26 | 0.41 |

[a]Data from the TPM, IRI, and PANAS are novel to this study. Data from LEAS and CTQ have previously been described [64].

[b]p-values are based on two-sample t-tests between those with high vs. low CTQ or CECA scores (based on median splits).

[c]Note that, after quality control checks, the final Ns in these analyses for the following measures were: IRI (N = 84) and PANAS (N = 84).

### Secondary analyses in low-empathy participants

Analogous results for analyses that took a median split on IRI Empathic Concern scores and focused on the low-Empathy group are reported in **Supplementary Materials**. These results were largely similar to those with high TPM scores. However, as can be seen in **Fig 4**, the positive relationship between LEAS Total scores and IRI Empathic Concern scores showed no difference in high vs. low CTQ Total scores.

## Discussion

In this study we addressed the question of whether psychopathic traits are positively or negatively associated with trait levels of emotional awareness (EA), and whether this relationship depends on the presence vs. absence of early adversity. Initial analyses across all participants found strong evidence for a negative relationship indicating that those with greater psychopathic traits showed lower levels of EA. Convergently, lower levels of empathic concern for others on a separate measure were also associated with lower EA. However, these relationships showed an interaction with early adversity levels. Specifically, the relationship between EA and psychopathic traits/low empathy was stronger in those with early adversity; and in some cases, the relationship in those without early adversity appeared largely absent. This suggested the presence of two distinct groups of individuals with high psychopathic traits in our sample–consistent with the idea that levels of secondary psychopathy (linked to early adversity) were more selectively associated with EA [30, 32].

In further support of the existence of heterogeneity within psychopathic profiles in our sample, we found that those with high psychopathy scores and early adversity reported greater negative affect than those without early adversity. Almost half of individuals with high psychopathy scores also showed levels of EA above the median value (and analyses using a separate measure also showed that about half of low-empathy individuals showed EA levels above the median value; see **Supplementary Materials**). Therefore, high psychopathy (and low empathy) did not prevent individuals from displaying high EA in general. However, further analyses in high-psychopathy individuals revealed that EA and psychopathy were strongly negatively correlated in those with high early adversity, whereas high-psychopathy individuals without early adversity showed weaker relationships between these variables. A somewhat similar pattern of results was also seen when analyzing a second measure of empathy levels. This suggested that the initially observed negative relationship between EA and psychopathy scores was driven primarily by those with early trauma and negative affect (i.e., more consistent with a secondary psychopathy profile).

Another interesting result of this study was that LEAS scores were associated with the meanness and disinhibition subscales of the TPM, but not with the boldness subscale. The meanness and disinhibition subscales largely reflect features associated with secondary psychopathy, such as aggression, impulsivity, irresponsibility, and oppositionality. In contrast, the boldness subscale reflects features more consistent with primary psychopathy, such as dominance, persuasiveness, social assurance, resiliency, and intrepidness. The selective lack of relationship between LEAS and boldness could therefore be seen as further evidence supporting the idea that diminished EA is not associated with a primary psychopathy profile. In addition, high-psychopathy individuals with high early adversity showed lower boldness than those with low early adversity, which also supports the presence of these distinct psychopathy profiles within the sample.

Jointly, these results therefore suggest that EA is selectively associated with secondary psychopathy levels. The weaker association between EA and psychopathy scores in those without early adversity (more consistent with primary psychopathy) could help account for these

individuals' prowess at displaying superficial charm, manipulating others, and attaining leadership roles (i.e., more closely associated with the TPM Boldness subscale, which was unrelated to EA). In contrast, the stronger association between EA and TPM scores in those with features linked to secondary psychopathy (i.e., early maltreatment and elevated negative affect) is consistent with the high impulsivity and high criminality seen in this population (i.e., more strongly linked to the TPM Meanness and Disinhibition subscales, which had significant associations with EA [7, 9, 12, 23, 26, 90–92]). This finding is also broadly consistent with previously observed relationships between psychopathy and alexithymia, emotion dysregulation, deficits in emotion recognition and theory of mind, and early adversity [13, 14, 19–22, 26, 28, 64, 93–95].

These considerations highlight some important limitations of the current study and point toward future directions. First, we did not collect measures of emotion recognition or cognitive theory of mind, which prevented us from assessing whether previously observed differences in these related abilities were also present in our sample. While not related to our primary hypotheses, such measures could have offered additional insights. It would be helpful for future studies to replicate our findings while also including such measures. Second, this study was conducted on a student sample and did not include violent or incarcerated offenders or other older and community-dwelling individuals that could include high-functioning psychopaths. Although the means for each TPM subscale in our sample were generally lower than those reported in previous forensic clinical or correctional populations [96–98], the range of scores was not dissimilar. For example, the range for our student sample's TPM Total score was 30–111, and the range for a prisoner sample was 25–119 [97]. While the student sample might not be representative of clinical or correctional populations, it does appear to capture the continuum of psychopathic traits that may be present in the general population, as the means were similar to those reported by others with student and community groups [97–100]. As our results may not generalize to the highest levels of psychopathy, it will be important to assess more severe populations in future studies. This also relates to limitations in our selective analyses of high-psychopathy individuals based on a median split. Namely, while those scoring above the median reflect higher levels within a student population, this may not capture high levels in a community or prison population. Other ways of dividing individuals into high and low groups were also possible.

Another limitation is that we chose to use a single measure of psychopathy, when several other measures exist, each with their strengths and limitations [101, 102]. Thus, we cannot be sure that our results would replicate using other psychopathy measures. Future research should therefore also seek to replicate these results with multiple additional measures of psychopathic traits. Additionally, our measurement of early adversity, while standard within existing literature, does not measure some other early environmental factors that could impact development, such as illness, poverty, environmental toxins, etc. Future research should explore these other aspects as they relate to psychopathy. Life history strategy is another construct that could be explored within the context of psychopathy and emotional awareness, as previous literature shows connections to both [64, 103]. Also, some measures collected for this study relied upon self-report questionnaires and were completed without supervision. Therefore, despite efforts to ensure data quality, inattentive completion of these measures cannot be ruled out. Finally, our sample size was only moderate, and it included more females than males. As TPM scores were higher in males, this raises the potential concern that the relationships we observed with TPM scores could be confounded by effects of sex. However, we included sex as a possible predictor in all analyses and the hypothesized relationships remained present. The general pattern of results also remained unchanged when looking at males and

females separately (see **S1 Fig in S1 File**). While these results reduce this concern, future studies should also confirm the effects we observed in a larger male sample.

In conclusion, this study represents an initial step in assessing the relationship between emotional awareness and psychopathic traits, as well as how it may depend on etiological factors previously associated with primary vs. secondary psychopathy [33]. Our results most strongly support a negative relationship between emotional awareness and a secondary psychopathy profile, based on observed interactions with early adversity and differences in negative affect. These results open up a number of additional questions that provide important avenues for future research–and that might offer additional insights about understanding and working with psychopathic populations. Specifically, they raise the possibility that interventions to improve EA, which have been demonstrated in other contexts, might be useful in the context of secondary psychopathy.

## Supporting information

**S1 File. This file contains all supplementary analyses and figures referred to in the main text.**
(PDF)

**S2 File. Study data.** This file includes all data used for the analyses reported in this study.
(CSV)

## Acknowledgments

The authors would like to thank Claire Lavalley for assistance in data processing.

## Author Contributions

**Conceptualization:** Ryan Smith, John J. B. Allen, Richard D. Lane.

**Data curation:** Anne E. Chuning, Colin A. Tidwell.

**Formal analysis:** Ryan Smith, Anne E. Chuning.

**Methodology:** Ryan Smith, Colin A. Tidwell.

**Resources:** John J. B. Allen.

**Supervision:** John J. B. Allen, Richard D. Lane.

**Visualization:** Ryan Smith.

**Writing – original draft:** Ryan Smith.

**Writing – review & editing:** Anne E. Chuning, Colin A. Tidwell, John J. B. Allen, Richard D. Lane.

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
