## [Decision Letter · Decision Letter 0]

25 Apr 2022

PONE-D-21-33144Psychopathic tendencies are selectively associated with reduced emotional awareness in the context of early adversityPLOS ONE

Dear Dr. Smith,

Thank you for submitting your manuscript to PLOS ONE. After careful consideration, we feel that it has merit but does not fully meet PLOS ONE’s publication criteria as it currently stands. Therefore, we invite you to submit a revised version of the manuscript that addresses the points raised during the review process.

We look forward to receiving your revised manuscript.

Kind regards,

Matthew S. Shane, PhD

Academic Editor

PLOS ONE

“R.S. is supported by the William K. Warren Foundation and the National Institute of General Medical Sciences (P20GM121312).”

“R.S. is supported by the William K. Warren Foundation (https://www.williamkwarrenfoundation.org/) and the National Institute of General Medical Sciences (P20GM121312; https://www.nigms.nih.gov/). The funders had no role in study design, data collection and analysis, decision to publish, or preparation of the manuscript.”

3. We noted in your submission details that a portion of your manuscript may have been presented or published elsewhere. [In a separate paper under review, we also use the data on emotional awareness and early adversity to answer a separate research question about how early adversity influences emotional awareness. We do not reproduce any of those analyses here and explicitly acknowledge this other paper in our manuscript. In this manuscript we instead examine how emotional awareness measures relate to psychopathy measures and empathy measures, and whether early adversity and negative affect moderate those relationships. The data on psychopathy, empathy, and negative affect has not been used in any previous study. A preprint of this other paper can be found here: https://psyarxiv.com/7nzqk/]  Please clarify whether this [conference proceeding or publication] was peer-reviewed and formally published. If this work was previously peer-reviewed and published, in the cover letter please provide the reason that this work does not constitute dual publication and should be included in the current manuscript.

4. We note that you have referenced (Patrick CJ. Operationalizing the triarchic conceptualization of psychopathy: Preliminary description of brief scales for assessment of boldness, meanness, and disinhibition. Unpublished test manual Tallahassee, FL,: Florida State University; 2010.which has currently not yet been accepted for publication. Please remove this from your References and amend this to state in the body of your manuscript: as detailed online in our guide for authors

http://journals.plos.org/plosone/s/submission-guidelines#loc-reference-style ".

Additional Editor Comments:

Thank you for submitting your top work to PLoS One for consideration.

First, an apology: Obtaining reviewers has been challenging during the pandemic, and I appreciate your patience throughout the process. At this point, I have been able to garner a high-quality review from one expert in the field, who has had the opportunity to provide detailed feedback on your submission. This reviewer has noted numerous strengths of the paper, including its high potential for theoretical import, and its overall statistical clarity. That said, the reviewer has also noted several features of the manuscript that decreased their enthusiasm somewhat. Overall, they have recommended a Major Revision.

I too have had the opportunity to review the manuscript, and am in broad agreement with the reviewer regarding the relative strengths and weaknesses of the manuscript. I believe the reviewer comments to be both clear and well-regarded, and so will not repeat them here. I will however expand somewhat on their concern regarding the method used to separate primary versus secondary psychopathy in the study. Whereas the reviewer has requested that you provide additional justification for these methods, I would request that you go one (or two) steps further than this, and also consider (regardless of your ability to find historical support for the method) its appropriateness in the present context. I do understand the logic – in fact, as someone who has recently been working with the PD-scale from the IRI, I quite like the logic of relating it to primary/secondary psychopathy concepts. However, relating it to primary/secondary psychopathy, and *defining it as* primary/secondary psychopathy are not the same thing. Moreover, in the present context, which is a study that relies on the reporting of relationships between different self-report measures, I wonder if your use of the PD scale in this manner doesn’t add substantive bias into the study design. You are, after all, predicting that psychopathy will be related to reduced emotional awareness, and are then using a measure of personal distress – which is established as a substantive correlate of emotional awareness – to demonstrate that only a subset of your psychopathy group shows this reduced EA. It all fits, it all makes sense…but is it in fact a somewhat circular process, wherein the category being used to separate participant groups is related to the dependent measure being evaluated for differences between groups (see Kriegerskorte et al., 2008)?

In addition, while I agree with the reviewer that your description of Bayes Factors was clear and helpful, I am nonetheless left wondering if those BF analyses were necessary. Perhaps some additional language that does more than simply explain how to interpret the BFs, but also informs on how they help further interpretation beyond the reported r and t-statistics (assuming they do) would be helpful.

In total, I agree with the reviewer that a Major Revision is in order, and welcome you to submit your revised manuscript for reconsideration at PLoS One. I do want to stress, however, that issues regarding potential circularity would need to be rectified before publication were possible; so I urge you to consider this issue, and its potential downstream empirical and theoretical impacts, carefully.

Best,

Matthew Shane

Action Editor, PLoS One

Reviewers' comments:

Reviewer's Responses to Questions

**Comments to the Author**

1. Is the manuscript technically sound, and do the data support the conclusions?

Reviewer #1: Yes

2. Has the statistical analysis been performed appropriately and rigorously? 

Reviewer #1: I Don't Know

3. Have the authors made all data underlying the findings in their manuscript fully available?

Reviewer #1: Yes

4. Is the manuscript presented in an intelligible fashion and written in standard English?

Reviewer #1: Yes

5. Review Comments to the Author

Reviewer #1: This manuscript reports results from a study examining the relationship between psychopathic traits (as measured by TriPM) and emotional awareness (EA; as measured by LEAS), including examination of “primary” and “secondary” psychopathy in relation to EA. The authors report their findings of apparent EA deficits in psychopathy, particularly in a subgroup of participants with high TriPM scores who reported experiencing early adversity (as measured by CECA and CTQ). The questions about empathy, EA, and psychopathy addressed by the research are interesting and important to furthering understanding of the relationship between those variables. Additionally, the authors did a good job of describing the Bayes factor analysis statistical methodology, providing sufficient information for readers less familiar with the BF method to understand the reported results. It was helpful to have access to the supplementary information provided by the authors, and I commend the authors on their clear delineation between a priori and post-hoc analyses. Despite the many strengths of this manuscript, there are some areas that warrant revisions, particularly adding support for certain methodological decisions and improving some aspects of the interpretation of the study results. Should the authors revise the document to reflect the feedback below, the manuscript would likely be suitable for publication and constitute a valuable addition to the published literature in this area.

Major Concerns

(1) Questionable definitional/methodological choices for assessing constructs of interest

A. The chosen way of defining and measuring “primary” vs. “secondary” psychopathy needs additional explanation/empirical support. Have other studies delineated the two subtypes of psychopathy in this way? Please include citations that support the chosen way of defining and measuring these constructs (e.g., Explain why using the CECA, CTQ, PANAS, and IRI personal distress subscale to distinguish between psychopathic subtypes is a valid method.).

i. The use of the IRI personal distress subscale as a “convergent measure of primary vs. secondary psychopathy” (p. 9) was odd and requires additional explanation/justification. As the personal distress subscale of the IRI is one of the affective/emotional empathy subscales, the authors will need to add to their justification and explanation for the hypothesis that this affective empathy subscale would be associated with the “secondary” form of psychopathy.

B. In the analyses looking at psychopathy subtypes, the authors included high psychopathy/high early adversity, high psychopathy/low-early adversity, and low psychopathy groups.

i. The choice to use median splits on the various measures to create the subgroups for analyses needs to be explained and supported with citations, including an explanation on how this choice may have impacted the results.

ii. The choice to not compare high and low early adversity in the “low psychopathy” group must be explained, including any impact this choice may have had on the results and interpretations of the findings. Further, the authors should include results of analyses showing whether the high and low psychopathy groups differed with regard to levels of experienced early adversity.

(2) Interpretation of study results and explanation in the Discussion could be stronger

A. The authors used the TriPM as their psychopathy measure, and I would have liked to see more discussion of the subscale-level results, as it appears that the results would be in-line with the authors’ hypotheses about intact EA associated with “successful psychopaths” or those adept in manipulation. Specifically, findings that the negative relationship between LEAS Total Score and TriPM psychopathy was driven by the Meanness and Disinhibition subscales (and not Boldness; p. 13), suggests that individuals high in Boldness (but lower on the other two subscales), might have intact EA. The discussion would benefit from more explanation of the TriPM subscale findings, particularly as they relate the authors’ hypotheses about “primary” and “secondary” psychopathy.

i. Including the TriPM subscales in the conceptualization of “primary” vs. “secondary” psychopathy in the introduction could help tie this together.

ii. This could be brought up in the Discussion as well. For example, the authors stated, “The weaker association with EA in primary psychopathy could help account for these individuals’ prowess at displaying superficial charm, manipulating others, and attaining leadership roles, while the stronger association in secondary psychopathy is consistent with the high impulsivity and high criminality seen in this population.” The authors would be able to make a stronger statement if they mentioned the potential role of Boldness scores in this distinction. Table 2 (p. 18) shows that the two “high psychopathy” groups differed at the subscale level, so highlighting this would make the discussion of the findings stronger.

B. There are a few places where non-significant results (with p > .05) are still interpreted, for example being described as “marginal” effects (e.g., p. 16). This type of questionable practice takes away from the manuscript as a whole, and is unnecessary, so use of such interpretations of the results in the manuscript should be reconsidered.

C. The discussion of the findings for “primary” vs. “secondary” psychopathy (p. 20) was confusing. Specifically, the authors mention that LEAS scores for individuals in the high psychopathy/high early adversity group did not significantly differ from those in the high psychopathy/low early adversity group, but then go on to discuss how their results show support for a difference in EA between the two groups. A clearer explanation of these findings is warranted.

Minor Concerns

(1) In the very first sentence of the introduction (p. 3), it says “Psychopathy is a personality trait characterized by...”. First, it would be more accurate to refer to psychopathy as a “constellation of traits” rather than just a singular trait (though that may have been a typing error in the manuscript). The definition of psychopathy as being “characterized by antisocial tendencies, such as cruelty, manipulation, grandiosity, callousness, and lack of empathy” (p. 3) is a bit questionable, particularly since you used the TriPM, which has reduced emphasis on the “antisociality/criminality” component of psychopathy (unlike the PCL-R).

(2) The mention of and emphasis on “dark triad” traits in the introduction (e.g., p. 3) was surprising and a bit confusing, seeing as the abstract and title focus on psychopathy. It seems that the arguments made in the introduction to set up the rationale behind the authors’ own study could have been made through sticking with the psychopathy literature. If the inclusion of “dark triad” traits is essential to the study, adding more explanation of the different constructs and how the psychopathy literature may differ from the “dark triad” literature would be helpful.

(3) The authors appear to use an “emotional empathy” definition of “empathy” throughout the manuscript; however, the explanation of this definition is lacking. Specifically, on page 4, it says “Despite deficits in empathy (or “affective theory of mind”; i.e., emotion contagion in response to others’ emotions) ...”. The terms of “emotional contagion,” “affective theory of mind,” and “empathic concern” all seem to be used interchangeably in this manuscript; however, the constructs are not necessarily considered equivalent in the literature. I would suggest clarifying your definitions of empathy by clearly laying out your definitions of both “cognitive empathy” and “affective empathy,” as you use both in the manuscript and study design/results interpretation.

(4) The authors cite study time constraints and their a priori hypotheses (p. 9) as the reason for not including the two cognitive empathy subscales from the IRI (Perspective Taking, Fantasy); however, this choice seems to require more explanation. Specifically, the authors made efforts to include measures for convergent validity for the psychopathy subtyping, so it was odd that they would leave the 14 items from the two cognitive empathy subscales of the IRI out. It seems like it would have been helpful to include these subscales, and the justification for not including these items seemed insufficient, particularly because the authors highlight this as a study limitation (p. 21).

(5) There appears to be a typing error on page 19, in the “Low Empathy Analyses” section, where it mentions “IRI Empathic Accuracy scores,” and I believe it is supposed to say “IRI Empathic Concern scores.”

6. PLOS authors have the option to publish the peer review history of their article (what does this mean?). If published, this will include your full peer review and any attached files.

Reviewer #1: No

---

## [Author Response · Author response to Decision Letter 0]

9 Jun 2022

[PLEASE SEE THE RESPONSES TO REVIEWERS DOCUMENT INCLUDED AS AN ATTACHMENT WITH THIS REVISION, WHICH IS FORMATTED FOR GREATER CLARITY THAN COULD BE PROVIDED IN THE CUT-AND-PASTE VERSION HERE]

Responses to reviewers

Editor Comments:

EDITOR: I too have had the opportunity to review the manuscript, and am in broad agreement with the reviewer regarding the relative strengths and weaknesses of the manuscript. I believe the reviewer comments to be both clear and well-regarded, and so will not repeat them here. I will however expand somewhat on their concern regarding the method used to separate primary versus secondary psychopathy in the study. Whereas the reviewer has requested that you provide additional justification for these methods, I would request that you go one (or two) steps further than this, and also consider (regardless of your ability to find historical support for the method) its appropriateness in the present context. I do understand the logic – in fact, as someone who has recently been working with the PD-scale from the IRI, I quite like the logic of relating it to primary/secondary psychopathy concepts. However, relating it to primary/secondary psychopathy, and *defining it as* primary/secondary psychopathy are not the same thing. 

Reply #1: Thank you, these are very helpful points. We agree that it needs to be clearer that personal distress scale score is not a rigorous means of identifying individuals with primary vs. secondary psychopathy. To help address this issue, we have added/edited language to stress this point. We have removed any language suggesting that secondary psychopathy is defined as having high negative affect scores or personal distress (although we have referenced work [e.g., see ref 76] that discusses secondary psychopathy as being primarily differentiated in terms of high anxiety due to early adversity). We also agree with the reviewer that the personal distress scale is not a straightforward indicator of secondary psychopathy (its correlations with our other measures suggest it is most related to a disposition toward high-anxiety and negative affect, with little-to-no relation to empathic concern [r=.14, p=.06, BF=.99]). We have now framed their theoretical motivation more appropriately. And we now examine personal distress mainly in the context of supplemental analyses for the interested reader. In general, the paper now focuses more on how the relationship between emotional awareness and psychopathy/empathic concern is dependent on early adversity. These are more cautiously referred to as being consistent with (but not definitive of) primary vs. secondary psychopathy. Here are a couple examples of updated wording:

Pg. 6 - “In this study we addressed the potential relationship between EA and psychopathy and hypothesized that an association would be present. However, based on the work reviewed above, we expected that this relationship could differ between those with vs. without early adversity (i.e., associated with secondary vs. primary psychopathy profiles, respectively; e.g., see (46)). Based on the common link to early learning, we hypothesized that EA would be negatively correlated with psychopathy scores in those with early adversity (i.e., consistent with secondary psychopathy). Given the mixed literature regarding emotional abilities in primary psychopathy, we predicted the presence of a relationship in those with high psychopathy scores without early adversity, but did not have a strong directional hypothesis.”

Pg. 22 - “This suggested that the initially observed negative relationship between EA and psychopathy scores was driven primarily by those with early trauma and negative affect (i.e., more consistent with a secondary psychopathy profile).”

EDITOR: Moreover, in the present context, which is a study that relies on the reporting of relationships between different self-report measures, I wonder if your use of the PD scale in this manner doesn’t add substantive bias into the study design. You are, after all, predicting that psychopathy will be related to reduced emotional awareness, and are then using a measure of personal distress – which is established as a substantive correlate of emotional awareness – to demonstrate that only a subset of your psychopathy group shows this reduced EA. It all fits, it all makes sense…but is it in fact a somewhat circular process, wherein the category being used to separate participant groups is related to the dependent measure being evaluated for differences between groups (see Kriegerskorte et al., 2008)?

Reply #2: This point about potential circularity is well-taken. It is true that some measures of emotional awareness (e.g., alexithymia using the self-reported TAS-20) are reliably correlated with distress and negative affect. This is one major reason we instead focus on the levels of emotional awareness scale (LEAS), which is often used in psychosomatic medicine research specifically because it is not confounded by personal distress and negative affect (i.e., while it does show associations with several medical/psychiatric conditions). This has been shown in several studies. Most importantly here, the LEAS is also not correlated with personal distress scores in our sample. As shown in Fig 3, LEAS shows a non-significant positive correlation (r=.12) with personal distress. So, if anything, it would bias results in the opposite direction. We think it is important to emphasize this point more, as it is a very natural concern to have. To do so, we have added/revised language to make this clearer. However, as described in Reply #1, we have also de-emphasized/re-framed some analyses using the PD scale that were the point of concern. Here are examples of how we’ve stressed the independence of LEAS and personal distress, and how we’ve tried to clarify the role of the PD scale in the analyses we have kept:

Pg. 14 - “Notably, LEAS scores were not significantly related to IRI personal distress scores, consistent with prior studies indicating that, unlike alexithymia measures (e.g., (74)), LEAS is independent of negative affect (75). Similarly, LEAS Total scores were also not related to PANAS negative affect scores (r = .09, p = .24, BF = .34) or positive affect scores (r = .1, p = .2, BF = .38).”

Pg. 22 - “In further support of the existence of heterogeneity within psychopathic profiles in our sample, we found that those with high psychopathy scores and early adversity reported greater negative affect than those without early adversity. In contrast, these two groups did not show significant differences in EA. As observed previously, LEAS scores were also independent of negative affect and distress (75).”

Pg. 16 - “For the interested reader, we provide detailed results of analyses (analogous to those above) for IRI personal distress within Supplementary Materials. There was a notable interaction between sex and LEAS Total scores in predicting IRI Personal Distress scores, which showed a positive relationship in males (r = .41, p = .009, BF = 6.86), but no relationship in females (r = .01, p = .92, BF = .20; see Fig 1). These analyses were considered exploratory, as we did not have strong a priori hypotheses regarding this measure. However, given the observed pattern of relationships between this measure and CTQ emotional abuse, TPM Boldness, and PANAS scores (see Fig 3), it might be viewed in this context as a convergent measure of dispositional negative affect (i.e., more consistent with the high-anxiety states associated with secondary psychopathy (76)).”

EDITOR: In addition, while I agree with the reviewer that your description of Bayes Factors was clear and helpful, I am nonetheless left wondering if those BF analyses were necessary. Perhaps some additional language that does more than simply explain how to interpret the BFs, but also informs on how they help further interpretation beyond the reported r and t-statistics (assuming they do) would be helpful.

Reply #3: We agree it could be helpful to make sure the benefits of the statistical approach are clear. One important benefit is interpretability. When only computing p-values within a frequentist approach, all one can do is either reject or fail to reject the null hypothesis. In contrast, the Bayesian approach evaluates the probability of the data under a null model as well as other models. This means you can calculate evidence for a null model as opposed to simply failing to reject it (e.g., a BF = 1/3 indicates 3 times more evidence for a null model than for an alternative model, while BF = 1 indicates equal evidence for null and alternative hypotheses). This is useful in the present case, since it indicates not only where tested relationships are likely present but also where they are likely absent. Another benefit is model comparison. By identifying the model with the most evidence, this provides a way of selecting which predictors should be included in subsequent analyses. We have added/revised some language as follows to clarify this:

Pg. 10 - “A BF represents the ratio of the probability of observed data under one model vs. another (i.e., where a higher probability of data under a model provides more evidence for that model). For example, BF = 1/3 indicates that data are three times more likely under the null model than the alternative model, BF = 1 indicates equal evidence for the null and alternative models, and BF = 3 indicates that the data are three times more probable under the alternative model than the null model. When interpreting the strength of evidence of each result, we adopt the guidelines described in Lee and Wagenmakers (71): BF = 1-3, poor/anecdotal evidence for the alternative hypothesis; 3-10, moderate evidence; 10-30, strong evidence; 30-100, very strong evidence; >100, extremely strong evidence. In cases of extremely strong evidence, we will indicate BF > 100 (as these values can get cumbersomely large under default priors in large sample sizes or when one model fits much better than another); we further compare BFs for the 1st- and 2nd-best model to offer a better sense of the relative evidence of the most competitive models. These BF analyses provide several advantages. Most importantly here, they provide a straightforward basis for model selection (i.e., indicating which predictors are relevant to include), and they allow the evaluation of evidence for the null model as well as for alterative models. Thus, they improve interpretability by allowing us to gain confidence that a relationship between variables is absent vs. present (i.e., as opposed to simply failing to reject a null hypothesis).”

We also make several statements in the manuscript indicating when our results provided evidence against the presence of specific relationships, which we think strengthens the contribution of the analyses.

Review Comments to the Author

Reviewer #1: This manuscript reports results from a study examining the relationship between psychopathic traits (as measured by TriPM) and emotional awareness (EA; as measured by LEAS), including examination of “primary” and “secondary” psychopathy in relation to EA. The authors report their findings of apparent EA deficits in psychopathy, particularly in a subgroup of participants with high TriPM scores who reported experiencing early adversity (as measured by CECA and CTQ). The questions about empathy, EA, and psychopathy addressed by the research are interesting and important to furthering understanding of the relationship between those variables. Additionally, the authors did a good job of describing the Bayes factor analysis statistical methodology, providing sufficient information for readers less familiar with the BF method to understand the reported results. It was helpful to have access to the supplementary information provided by the authors, and I commend the authors on their clear delineation between a priori and post-hoc analyses. Despite the many strengths of this manuscript, there are some areas that warrant revisions, particularly adding support for certain methodological decisions and improving some aspects of the interpretation of the study results. Should the authors revise the document to reflect the feedback below, the manuscript would likely be suitable for publication and constitute a valuable addition to the published literature in this area.

Major Concerns

(1) Questionable definitional/methodological choices for assessing constructs of interest

A. The chosen way of defining and measuring “primary” vs. “secondary” psychopathy needs additional explanation/empirical support. Have other studies delineated the two subtypes of psychopathy in this way? Please include citations that support the chosen way of defining and measuring these constructs (e.g., Explain why using the CECA, CTQ, PANAS, and IRI personal distress subscale to distinguish between psychopathic subtypes is a valid method.).

Reply #4: We agree with the reviewer that it was not appropriate to define primary/secondary psychopathy based on early adversity and negative affect measures. Indeed, many papers evaluate primary vs. secondary psychopathy based on additional psychopathy features (although, high-anxiety in response to early adversity is often discussed as a central characteristic for distinguishing secondary psychopathy; e.g., see ref 76 in the manuscript). As such, we have revised language to make it clear that early adversity and negative affect are only features consistent with primary vs. secondary psychopathy, and are not definitive. We provided a few quotes of updated language in Reply #1 and #2 to exemplify these revisions.

i. The use of the IRI personal distress subscale as a “convergent measure of primary vs. secondary psychopathy” (p. 9) was odd and requires additional explanation/justification. As the personal distress subscale of the IRI is one of the affective/emotional empathy subscales, the authors will need to add to their justification and explanation for the hypothesis that this affective empathy subscale would be associated with the “secondary” form of psychopathy.

Reply #5: As describe in Replies #1 and #2, we agree that interpreting the personal distress subscale as a convergent measure of secondary psychopathy is not straightforward. We have therefore removed this framing/discussion of this subscale. It is now primarily used to provide secondary insights for the interested reader in supplementary materials. However, we also added some language indicating – based on its correlations with other measures and past characterizations in previous studies – why it might be seen as reflecting dispositions toward the high-anxiety states associated with secondary psychopathy. But we make no claims that personal distress is a strong indicator of secondary psychopathy.

Pg. 16 - “…given the observed pattern of relationships between this measure and CTQ emotional abuse, TPM Boldness, and PANAS scores (see Fig 3), it might be viewed in this context as a convergent measure of dispositional negative affect (i.e., more consistent with the high-anxiety states associated with secondary psychopathy (76)).”

Note that this section references Fig 3, which shows that distress is higher with emotional abuse, lower with TPM Boldness, higher with greater negative affect, and lower with greater positive affect. This makes sense if you look at the specific items of the scale, which do not reflect concern for others, but simply the tendency to enter high-anxiety states in situations where harm is present.

B. In the analyses looking at psychopathy subtypes, the authors included high psychopathy/high early adversity, high psychopathy/low-early adversity, and low psychopathy groups.

i. The choice to use median splits on the various measures to create the subgroups for analyses needs to be explained and supported with citations, including an explanation on how this choice may have impacted the results.

ii. The choice to not compare high and low early adversity in the “low psychopathy” group must be explained, including any impact this choice may have had on the results and interpretations of the findings. Further, the authors should include results of analyses showing whether the high and low psychopathy groups differed with regard to levels of experienced early adversity.

Reply #6: We thank the reviewer for highlighting these helpful points. One thing that should be better emphasized/explained in the manuscript is that most median split analyses were done largely for secondary interpretive/visualization purposes. Our primary results assessed the interaction between LEAS and early adversity (CTQ/CECA) in predicting psychopathy scores. The significant interaction found in our models indicated a stronger negative relationship between LEAS and psychopathy scores in those with greater early adversity. However, as these are all continuous measures, this interaction is hard to illustrate. Our solution was to show correlations using the median splits as a way to visualize how the relationship between EA and psychopathy differed in the high vs. low early adversity groups. Most analyses performed on the median split groups were therefore secondary. We are not aware of specific citations to provide for previous work that has used median splits on the TPM for similar purposes. 

In some other secondary analyses, we used a median split to selectively examine individuals with high psychopathy scores. These were also done largely for purposes of interpreting our primary results, by confirming whether the relationship to EA still depended on early adversity when only looking at those high in psychopathic traits. This is the case the reviewer refers to when stating that we divided individuals into 3 groups (i.e., low psychopathy, and high psychopathy with vs. without early adversity). However, it is more appropriate to say that we selectively examined the two high-psychopathy groups, and that the low-psychopathy participants were not the focus of these analyses. In other words, we were trying to answer a question about heterogeneity within those high in psychopathic traits, so the low-psychopathy participants were not relevant to this question.

To make this clearer, we have adjusted language in several places to make sure it is clear that: (1) median splits were used either simply for visualization or for interpretation of our primary results, and (2) low-psychopathy participants were not relevant to questions about heterogeneity within psychopathic populations, and were therefore not the focus of those analyses. We also added in the requested analyses of early adversity differences in high vs. low psychopathy groups (and for low vs. high empathy groups in supplementary materials). We also acknowledged potential limitations of the choice to use median splits in the discussion. Here are a couple examples of relevant language in the revised manuscript:

Pg. 13 - “The interaction between LEAS and CECA scores was driven by a stronger negative relationship between LEAS scores and TPM scores in those with higher CECA scores. This could be visualized by taking a median split on CECA scores (low < 14.75, high ≥ 14.75) and correlating LEAS and TPM scores for those with high vs. low CECA scores separately (see Fig 2). Although displayed here primarily for purposes of visualization, we note that the correlation in those with high CECA scores was r = -.42 (p < .001, BF > 100), while the correlation in those with low CECA scores was r = -.12 (p = .29, BF = .42). These correlations were also significantly different (z = -2.14, p = .03).”

Pg. 17 - “To more thoroughly interpret the significant interactions between psychopathy, early adversity, and emotional awareness identified in our primary analyses above, we performed additional post-hoc analyses to selectively probe these relationships in participants with high psychopathy scores. To do so, we performed a median split on TPM scores (low < 59, high ≥ 59) and then restricted analyses to high-TPM participants. We then used the same analysis approach as above to test if the interaction between LEAS scores and early adversity scores was still present in predicting psychopathy scores. This allowed us to test for evidence of heterogeneity in EA within high-psychopathy individuals in relation to early adversity (i.e., consistent with primary vs. secondary psychopathy). Note that our question here was specifically about the possibility of distinct patterns in individuals showing high psychopathy scores; thus, the low-psychopathy individuals (below the median for TPM scores; 10/40 males, 78/137 females) were not included in these analyses.

As in our main analyses above, the winning models still included this interaction… 

As above, for further interpretation and visualization of these interactions (see Fig 4), the high-psychopathy individuals were then divided into groups with low vs. high levels of early adversity, which allowed additional examination of EA in those with profiles more consistent with primary vs. secondary psychopathy. To do so, we performed a median split on CTQ (low < 6.67, high ≥ 6.67) and CECA (low < 16.5, high ≥ 16.5) Total scores. We then divided participants into two high-TPM groups for each measure: high-TPM/high-early adversity (CTQ: 10/40 males, 33/137 females; CECA: 14/40 males, 31/137 females) and high-TPM/low-early adversity (CTQ: 20/40 males, 26/137 females; CECA: 16/40 males, 28/137 females). 

Illustrating the significant interactions described above, Fig 4 shows that those with high psychopathy and high early adversity showed significant negative relationships between TPM Total scores and LEAS Total Scores…” 

Pg. 18 - “Comparison of high- and low-psychopathy individuals also confirmed that high-psychopathy individuals had experienced greater early adversity overall with respect to CECA Total scores (t(175) = 3.73, p < .001, d = 0.56, BF = 88.73) and CTQ Total scores (t(175) = 2.93, p = .004, d = 0.44, BF = 8.3).”

Pg. 24 - “This also relates to limitations in our selective analyses of high-psychopathy individuals based on a median split. Namely, while those scoring above the median reflect higher levels within a student population, this may not capture high levels in a community or prison population. Other ways of dividing individuals into high and low groups were also possible.”

(2) Interpretation of study results and explanation in the Discussion could be stronger

A. The authors used the TriPM as their psychopathy measure, and I would have liked to see more discussion of the subscale-level results, as it appears that the results would be in-line with the authors’ hypotheses about intact EA associated with “successful psychopaths” or those adept in manipulation. Specifically, findings that the negative relationship between LEAS Total Score and TriPM psychopathy was driven by the Meanness and Disinhibition subscales (and not Boldness; p. 13), suggests that individuals high in Boldness (but lower on the other two subscales), might have intact EA. The discussion would benefit from more explanation of the TriPM subscale findings, particularly as they relate the authors’ hypotheses about “primary” and “secondary” psychopathy.

i. Including the TriPM subscales in the conceptualization of “primary” vs. “secondary” psychopathy in the introduction could help tie this together.

ii. This could be brought up in the Discussion as well. For example, the authors stated, “The weaker association with EA in primary psychopathy could help account for these individuals’ prowess at displaying superficial charm, manipulating others, and attaining leadership roles, while the stronger association in secondary psychopathy is consistent with the high impulsivity and high criminality seen in this population.” The authors would be able to make a stronger statement if they mentioned the potential role of Boldness scores in this distinction. Table 2 (p. 18) shows that the two “high psychopathy” groups differed at the subscale level, so highlighting this would make the discussion of the findings stronger.

Reply #7: This is a very good point, which we had not thoroughly considered. We have now added discussion of this, as suggested. For example:

Pg. 22 - “Another interesting result of this study was that LEAS scores were associated with the meanness and disinhibition subscales of the TPM, but not with the boldness subscale. The meanness and disinhibition subscales largely reflect features associated with secondary psychopathy, such as aggression, impulsivity, irresponsibility, and oppositionality. In contrast, the boldness subscale reflects features more consistent with primary psychopathy, such as dominance, persuasiveness, social assurance, resiliency, and intrepidness. The selective lack of relationship between LEAS and boldness could therefore be seen as further evidence supporting the idea that EA is not associated with a primary psychopathy profile. In addition, high-psychopathy individuals with high early adversity showed lower boldness than those with low early adversity, which also supports the presence of distinct psychopathy profiles within the sample.”

B. There are a few places where non-significant results (with p > .05) are still interpreted, for example being described as “marginal” effects (e.g., p. 16). This type of questionable practice takes away from the manuscript as a whole, and is unnecessary, so use of such interpretations of the results in the manuscript should be reconsidered.

Reply #8: We appreciate this point. In all cases, we have now removed statements about marginal results and simply state that these results are non-significant.

C. The discussion of the findings for “primary” vs. “secondary” psychopathy (p. 20) was confusing. Specifically, the authors mention that LEAS scores for individuals in the high psychopathy/high early adversity group did not significantly differ from those in the high psychopathy/low early adversity group, but then go on to discuss how their results show support for a difference in EA between the two groups. A clearer explanation of these findings is warranted.

Reply #9: We apologize for the lack of clarity. In the revised manuscript, we have revised wording to make the relevant points clearer. The important point is that only the early adversity group shows a strong relationship between psychopathy scores and emotional awareness. So the groups don’t show differences in mean levels of EA. Instead, the group difference we are discussing is meant to highlight that the level of psychopathy is only related to EA in the high early adversity participants. Throughout the discussion, we have taken care to always say the difference is in these relationships, and not that there is a group difference in mean levels of EA. Here is an example revised paragraph:

Pg. 22 - “In further support of the existence of heterogeneity within psychopathic profiles in our sample, we found that those with high psychopathy scores and early adversity reported greater negative affect than those without early adversity. In contrast, these two groups did not show significant differences in EA. As observed previously, LEAS scores were also independent of negative affect and distress (75). Roughly one third of individuals with high psychopathy scores also showed levels of EA above the median value (and analyses using a separate measure also showed that over one third of low-empathy individuals showed EA levels above the median value; see Supplementary Materials). Therefore, high psychopathy (and low empathy) did not prevent individuals from displaying high EA in general. However, further analyses in high-psychopathy individuals revealed that EA and psychopathy were strongly negatively correlated in those with high early adversity, whereas high-psychopathy individuals without early adversity showed little-to-no relationship between these variables. A somewhat similar pattern of results was also seen when analyzing a second measure of empathy levels. This suggested that the initially observed negative relationship between EA and psychopathy scores was driven primarily by those with early trauma and negative affect (i.e., more consistent with a secondary psychopathy profile).”

Minor Concerns

(1) In the very first sentence of the introduction (p. 3), it says “Psychopathy is a personality trait characterized by...”. First, it would be more accurate to refer to psychopathy as a “constellation of traits” rather than just a singular trait (though that may have been a typing error in the manuscript). The definition of psychopathy as being “characterized by antisocial tendencies, such as cruelty, manipulation, grandiosity, callousness, and lack of empathy” (p. 3) is a bit questionable, particularly since you used the TriPM, which has reduced emphasis on the “antisociality/criminality” component of psychopathy (unlike the PCL-R).

Reply #10: We have now revised the wording to refer to a “constellation of antisocial personality traits”, along the lines of the reviewers’ suggestion. Otherwise, our revised opening description of psychopathic traits simply aims to list the range of dispositions associated with the general construct of psychopathy. In our view, this is appropriate as a broad introductory description, where the focus then becomes more specific as the paper progresses. If the editor feels this introductory description requires further editing, we are happy to consider this in a subsequent revision. The revised version states:

Pg. 3 - “Psychopathy refers to a constellation of antisocial personality traits, such as cruelty, manipulativeness, grandiosity, callousness, and lack of empathy (1). Individuals with psychopathy are also characterized as being high in egocentrism/narcissism, exploitativeness, impulsivity, aggression, and criminality, while showing low remorse and fear, and shallow affect more generally (2, 3).”

(2) The mention of and emphasis on “dark triad” traits in the introduction (e.g., p. 3) was surprising and a bit confusing, seeing as the abstract and title focus on psychopathy. It seems that the arguments made in the introduction to set up the rationale behind the authors’ own study could have been made through sticking with the psychopathy literature. If the inclusion of “dark triad” traits is essential to the study, adding more explanation of the different constructs and how the psychopathy literature may differ from the “dark triad” literature would be helpful.

Reply #11: We agree that explicit description of the dark triad concept may have distracted from the main focus. The dark triad mainly divides the broader set of traits associated with psychopathy into three constructs that also include Machiavellianism and Narcissism. We felt this literature should not be ignored in our review (as it includes many studies of psychopathic traits), but also agree that it may not be helpful to discuss the separate triad constructs in detail. To ameliorate this, we have removed explicit discussion of the dark triad and kept the focus on the broader psychopathy construct. However, we still cite some of the dark triad literature when it offers relevant results in relation to psychopathy more broadly.

(3) The authors appear to use an “emotional empathy” definition of “empathy” throughout the manuscript; however, the explanation of this definition is lacking. Specifically, on page 4, it says “Despite deficits in empathy (or “affective theory of mind”; i.e., emotion contagion in response to others’ emotions) ...”. The terms of “emotional contagion,” “affective theory of mind,” and “empathic concern” all seem to be used interchangeably in this manuscript; however, the constructs are not necessarily considered equivalent in the literature. I would suggest clarifying your definitions of empathy by clearly laying out your definitions of both “cognitive empathy” and “affective empathy,” as you use both in the manuscript and study design/results interpretation.

Reply #13: We appreciate this suggestion. It is true that the literature does not always use these terms consistently. Some papers we cite use “cognitive empathy” in a way that is synonymous with cognitive theory of mind (as applied to others’ emotions). Others use “affective theory of mind” in a manner similar to “emotional” empathy (i.e., caring that another person is sad and being motivated to make them feel better – where in some sense it “hurts” to see others sad, which relates to a type of emotion contagion). Our choice of terms was driven mainly by consideration of how these terms were used in the papers we cited. But we agree it could be helpful to revise the wording in hopes of making definitions a bit clearer. As such, we have revised the section as follows:

Pg. 3 - “One common question pertains to the relationship between psychopathy and socio-emotional abilities. On the one hand, empathy deficits in psychopathy (i.e., lack of concern for others) might be expected to hinder such abilities. On the other hand, the manipulation skills, superficial charm, and successful attainment of leadership roles in psychopathy each suggest a proficiency in detecting and capitalizing on the emotions of others (7, 9, 11, 12). Individuals with psychopathic traits have also been purported to use empathy as a manipulative strategy (13-15). However, the literature on this topic is mixed. For example, while psychopathy is associated with a reduced tendency to feel negative emotion in response to the suffering of others (i.e., affective empathy; sometimes also called affective theory of mind), some studies in those with psychopathic traits have observed intact abilities to recognize the thoughts or feelings of others (i.e., cognitive theory of mind or cognitive empathy; e.g., see (16-18)). Yet, other studies find that psychopathic traits are negatively correlated with these cognitive abilities (13, 14, 19-22). Studies examining emotional intelligence (EI) also find mixed results, with intact EI present in some studies (and/or assessment measures) but not others (e.g., (23-25)).”

(4) The authors cite study time constraints and their a priori hypotheses (p. 9) as the reason for not including the two cognitive empathy subscales from the IRI (Perspective Taking, Fantasy); however, this choice seems to require more explanation. Specifically, the authors made efforts to include measures for convergent validity for the psychopathy subtyping, so it was odd that they would leave the 14 items from the two cognitive empathy subscales of the IRI out. It seems like it would have been helpful to include these subscales, and the justification for not including these items seemed insufficient, particularly because the authors highlight this as a study limitation (p. 21).

Reply #14: We agree that including the cognitive subscales may have offered some interesting secondary insights. But these were not directly related to the hypothesis being tested. Our main hypothesis was about the relationship between EA and psychopathy (using TPM scores). As a secondary analyses, we were interested in a convergent measure of the affective empathy deficit in psychopathy, due to its particular relevance to our question about EA. The IRI empathic accuracy subscale seemed like a good tool for this purpose, and we collected personal distress as well for largely exploratory purposes. If we had more time, we would have liked to include the cognitive subscales, but time constraints required compromise here. Since our main hypothesis was not about cognitive theory of mind abilities in psychopathy, we judged these scales were the least problematic to remove to remain within our study time limits. It was mentioned as a limitation simply because it could have helped address other secondary questions of interest. But not because it necessarily limited how we could interpret our results themselves. With this in mind, we have made some revisions to language in a couple places in hopes of making it clear that they could offer secondary insights but were not central to our stated hypothesis. For example:

Pg. 9 - “The IRI (67) assesses both cognitive and affective components of dispositional empathy (2 subscales each). Based on our a priori hypotheses, and the study’s time constraints, we only assessed the two affective subscales: Empathic Concern (having sympathy for others in need) and Personal Distress (having negative arousal in response to perceived distress in others). While the two cognitive subscales had the potential to provide secondary insights, they were not prioritized because they were not required to assess the hypothesized relationship between EA, early adversity, and psychopathy/low affective empathy.”

Pg. 23-24 - “These considerations highlight some important limitations of the current study and point toward future directions. First, we did not collect measures of emotion recognition or cognitive theory of mind, which prevented us from assessing whether previously observed differences in these related abilities were also present in our sample. While not related to our primary hypotheses, such measures could have offered additional insights. It would be helpful for future studies to replicate our findings while also including such measures, so that potential inconsistencies with previous work can be addressed.”

(5) There appears to be a typing error on page 19, in the “Low Empathy Analyses” section, where it mentions “IRI Empathic Accuracy scores,” and I believe it is supposed to say “IRI Empathic Concern scores.”

Reply #15: Thanks for pointing out this typo. It has now been fixed.

Editorial office comments:

Reply #16: We have now ensured that manuscript adheres to the style requirements.

“R.S. is supported by the William K. Warren Foundation and the National Institute of General Medical Sciences (P20GM121312).”

“R.S. is supported by the William K. Warren Foundation (https://www.williamkwarrenfoundation.org/) and the National Institute of General Medical Sciences (P20GM121312; https://www.nigms.nih.gov/). The funders had no role in study design, data collection and analysis, decision to publish, or preparation of the manuscript.”

Reply #17: We have removed the funding statement from the manuscript and now provided these details in the cover letter, as requested.

3. We noted in your submission details that a portion of your manuscript may have been presented or published elsewhere. [In a separate paper under review, we also use the data on emotional awareness and early adversity to answer a separate research question about how early adversity influences emotional awareness. We do not reproduce any of those analyses here and explicitly acknowledge this other paper in our manuscript. In this manuscript we instead examine how emotional awareness measures relate to psychopathy measures and empathy measures, and whether early adversity and negative affect moderate those relationships. The data on psychopathy, empathy, and negative affect has not been used in any previous study. A preprint of this other paper can be found here: https://psyarxiv.com/7nzqk/] Please clarify whether this [conference proceeding or publication] was peer-reviewed and formally published. If this work was previously peer-reviewed and published, in the cover letter please provide the reason that this work does not constitute dual publication and should be included in the current manuscript.

Reply #18: This paper has now been published. The citation has now been included in the manuscript. It does not constitute dual publication for the reasons mentioned above. Namely, we use several measures not used in the previous paper, and all analyses/results reported in this paper are novel and were not included in the previous paper. We have now explained this in the cover letter.

4. We note that you have referenced (Patrick CJ. Operationalizing the triarchic conceptualization of psychopathy: Preliminary description of brief scales for assessment of boldness, meanness, and disinhibition. Unpublished test manual Tallahassee, FL,: Florida State University; 2010.which has currently not yet been accepted for publication. Please remove this from your References and amend this to state in the body of your manuscript: as detailed online in our guide for authors

http://journals.plos.org/plosone/s/submission-guidelines#loc-reference-style ".

Reply #19: We have now replaced this citation with a published paper covering the same topic.

---

## [Decision Letter · Decision Letter 1]

14 Aug 2022

PONE-D-21-33144R1Psychopathic tendencies are selectively associated with reduced emotional awareness in the context of early adversityPLOS ONE

Dear Dr. Smith,

Thank you for submitting your manuscript to PLOS ONE. After careful consideration, we feel that it has merit but does not fully meet PLOS ONE’s publication criteria as it currently stands. Therefore, we invite you to submit a revised version of the manuscript that addresses the points raised during the review process.

We look forward to receiving your revised manuscript.

Kind regards,

Peter Karl Jonason

Academic Editor

PLOS ONE

Reviewers' comments:

Reviewer's Responses to Questions

**Comments to the Author**

1. If the authors have adequately addressed your comments raised in a previous round of review and you feel that this manuscript is now acceptable for publication, you may indicate that here to bypass the “Comments to the Author” section, enter your conflict of interest statement in the “Confidential to Editor” section, and submit your "Accept" recommendation.

Reviewer #1: (No Response)

Reviewer #2: (No Response)

2. Is the manuscript technically sound, and do the data support the conclusions?

Reviewer #1: Yes

Reviewer #2: Yes

3. Has the statistical analysis been performed appropriately and rigorously? 

Reviewer #1: I Don't Know

Reviewer #2: I Don't Know

4. Have the authors made all data underlying the findings in their manuscript fully available?

Reviewer #1: Yes

Reviewer #2: Yes

5. Is the manuscript presented in an intelligible fashion and written in standard English?

Reviewer #1: Yes

Reviewer #2: Yes

6. Review Comments to the Author

Reviewer #1: This manuscript reports results from a study examining the relationship between psychopathic traits (as measured by TriPM) and emotional awareness (EA; as measured by LEAS), including examination of “primary” and “secondary” psychopathy profiles as they may relate to EA. The authors report their findings of apparent EA deficits in psychopathy, particularly in a subgroup of participants with high TriPM scores who reported experiencing early adversity (as measured by CECA and CTQ). The questions about empathy, EA, and psychopathy addressed by the research are interesting and important to furthering understanding of the relationship between those variables. The authors did a great job of addressing the concerns raised during the initial review through edits to the manuscript. Aside from some minor concerns, outlined below, I believe the manuscript is suitable for publication and will constitute a valuable addition to the published literature in this area.

Minor Concerns:

1) In the description of the various measures used (starting on page 7), I noticed that Cronbach’s alpha is included for the LEAS measure, but not for the other measures. The authors might consider adding Cronbach’s alpha values for the subscales of the TriPM, and the other measures, as possible and appropriate.

2) There appears to be a typing error on page 21, in the “Secondary analyses in low-empathy participants” section, where it mentions “IRI Empathic Accuracy scores,” and I believe it is supposed to say “IRI Empathic Concern scores.”

3) On page 22, in the Discussion section, the following sentence presented a bit of confusion: “In contrast, these two groups did not show significant differences in EA.”

(a) When reading this sentence, I interpreted the two groups to be the high-TPM/high-early adversity and high-TPM/low-early adversity based on the preceding sentence that mentions heterogenous psychopathic profiles in the sample. If this is a proper interpretation, the sentence expressing that there was no significant difference in EA between the groups is confusing given other statements that seem to suggest that the two groups did differ in EA, with the high-TPM/high-early adversity group having lower EA (lower LEAS scores) and the high-TPM/low-early adversity group not showing the same “impairment” in EA. For example, in the abstract, the final sentence reads: “This suggests that EA may be selectively associated with levels of secondary psychopathy, while those with high levels of primary psychopathy remain capable of high EA.” This sentence in the abstract seems at odds with the idea that there was no difference in EA between the groups being referenced in the above-mentioned sentence on page 22.

(b) It is possible my interpretation of this sentence or the results is incorrect, though I think some clarification would be helpful to avoid similar confusion in others who will read this.

4) On page 23 of the Discussion section, the following sentence is included: “The selective lack of relationship between LEAS and boldness could therefore be seen as further evidence supporting the idea that EA is not associated with a primary psychopathy profile.”

(a) The authors could clarify this sentence by specifying that “diminished EA” (or “impaired EA” or some other synonym) does not seem to be associated with a primary psychopathy profile.

(b) For example, the sentence could be worded like this: “The selective lack of relationship between LEAS and boldness could therefore be seen as further evidence supporting the idea that diminished EA is not associated with a primary psychopathy profile.”

5) Additionally, on page 23 of the Discussion section, the following is included and seemed a bit confusing: “However, it appears less consistent with work showing that individuals with primary psychopathy, but not secondary psychopathy, show emotion recognition deficits [15, 46] or with studies showing preserved “cognitive” theory of mind abilities for emotions of others [16-18].”

(a) The point of confusion is related to the second part of the above sentence. Specifically, it is unclear to me how the results of the present study are not consistent with studies showing preserved cognitive empathy/cognitive theory of mind in primary psychopathy.

(b) I would suggest removing this part of the sentence or clarifying how the results are inconsistent with those referenced previous study findings on cognitive empathy in primary psychopathy.

Reviewer #2: The authors address an interesting issue. The question of how adverse early experiences may contribute to adult psychopathology is an important one. The paper is generally well-written. However, along with these strengths, there are some weaknesses. For example, the sample is relatively small and the methodology is a somewhat roundabout way to address the question – using childhood adversity and lower empathy to signal secondary psychopathy rather than addressing the sub-facets of psychopathy directly. In what follows, page numbers refer to the clean document from the reviewing PDF. I do not object, in principle, to the authors analyzing data from the same participants who contributed to a different publication on a different hypothesis, but milking data from a sample of 177 participants into two papers does feel a bit exploitative especially when the three measures included in the previous publication were analyzed again here.

Just because there is an association whereby those with lower LEAS scores had higher TPM and lower IRI scores, does not suggest that those with lower TPM and higher IRI scores had HIGH LEAS scores per se – Associations indicate how relatively higher or lower scores covary but do not indicate whether scores are high relative to average scores, for example. So, the authors might think about rephrasing the last sentence of the abstract. It should also be clear throughout that the sample is not a clinical sample.

On p. 4, the authors contrast primary and secondary psychopathy by referring to insufficient arousal to emotional cues as being emblematic of primary psychopathy and referring to secondary psychopathy as stemming from early adversity. This is a contrast of inequivalences of sorts. The authors should focus on either early etiology or associations with emotion awareness rather than comparing apples to oranges.

We can’t know if secondary psychopaths are incarcerated at higher rates without knowing the prevalence of both types both in the general population and in prison populations. All we can know is whether a higher rate of tested prisoners are characterized as secondary rather than primary psychopaths. The authors still need to be more precise in their language. For example, on p. 5 (end of first paragraph), they mention the etiology of psychopathic traits. Do we know that adverse experiences contribute to the development of such tendencies, or do they just covary? Authors have to be careful when making causal assumptions based on correlations. It is possible that, if there is a genetic component, parents high in psychopathy are both more likely to abuse their children and pass on psychopathic traits, meaning that maltreatment itself is not an etiological factor. All of these ideas are interesting and important, but require a great deal of care in how they are described.

P. 5 at reference [44], less accurate compared to what? The authors should specify when making comparisons.

P. 5 – there should be citations following the statement that trait EA is widely recognized…

The hypothesis at the bottom of p. 5 (that there are different reasons to suspect both high or low emotional awareness in individuals with psychopathy and other dark traits) has been suggested by others (e.g., Vonk et al., 2013 with regard to narcissism, Esperger & Bereczkei, 2012; Nagler et al., 2014; Vonk et al., 2015). It might be helpful to frame in terms of the distinction between ability to read emotion states (cognitive empathy) and the capacity to feel for others (emotional or affective empathy). It is not clear to me whether EA is capturing the latter or a form of emotional intelligence and how this should relate to theory of mind and empathy more broadly. Because much of the prior literature is coached in terms of this distinction, it would help for these authors to clarify the relations among these various related constructs. For example, they mention using the subscales of the IRI as a convergent measure of empathy but is LEAS really a measure of empathy? I am not personally familiar with the LEAS. Suddenly, the PANAS is mentioned as a measure of affect in the Method section but it is unclear why each construct is included. It would help to see a formalization of the conceptual model proposed by the authors before the Method. The authors should also spend some time indicating how the construct of psychopathy is captured by the TPM compared to other measures that have been used in previous studies. Do the authors care about the overall TPM score or the subscales? Do the subscales represent primary and secondary psychopathy? If not, how do they fit into the conceptual model for the study? Near the bottom of p. 9, is psychopathy/low affective empathy meant to treat these two as synonymous? It is difficult to evaluate the statistical approach without greater clarity from the authors on the model.

In addition, the measures of childhood adversity appear to focus on a somewhat narrow range of experiences related to abuse rather than more broadly capturing things like poverty, exposure to dangerous environments, illness etc.

If the scoring for the LEAS came from a computer scoring method, where does the inter-rater reliability come from?

The authors should report reliability for all measures.

P. 9, why do the authors say “the empathic concern subscale acted as a convergent measure of low affective empathy in addition to the TPM”? The TPM doesn’t measure empathy???!!

I don’t find the rationale for the personal distress scale to be compelling as described on p. 9-10.

Confidence intervals should be reported for t-tests.

How was the sample size determined? Is the student sample even representative and valid for the questions posed? The authors address this issue in the discussion but it is a serious limitation of the research.

One of the most interesting findings is that “high psychopathy (and low empathy) did not prevent individuals from displaying high EA in general.” (p. 22). I think the paper could be more strongly framed as an investigation of whether the association between psychopathy and EA is moderated by early adversity and why this is the case. Although I can see why TPM scores are treated as the outcome, it seems it might make sense to run reverse regression models with TPM predicting LEAS, including adverse experiences as a moderator. I don’t agree with the authors’ focus on the distinction between primary and secondary psychopathy since these facets were not directly assessed.

The authors should report the correlations between measures rather than depicting this information graphically.

Were there any attention checks? How was the data screened/cleaned?

7. PLOS authors have the option to publish the peer review history of their article (what does this mean?). If published, this will include your full peer review and any attached files.

Reviewer #1: No

Reviewer #2: No

---

## [Author Response · Author response to Decision Letter 1]

29 Sep 2022

Please see our responses to reviewer comments at the end of the submission PDF (final attachment).

---

## [Decision Letter · Decision Letter 2]

28 Oct 2022

Psychopathic tendencies are selectively associated with reduced emotional awareness in the context of early adversity

PONE-D-21-33144R2

Dear Dr. Smith,

We’re pleased to inform you that your manuscript has been judged scientifically suitable for publication and will be formally accepted for publication once it meets all outstanding technical requirements.

Kind regards,

Peter Karl Jonason

Academic Editor

PLOS ONE

Additional Editor Comments (optional):

Reviewers' comments:

Reviewer's Responses to Questions

**Comments to the Author**

1. If the authors have adequately addressed your comments raised in a previous round of review and you feel that this manuscript is now acceptable for publication, you may indicate that here to bypass the “Comments to the Author” section, enter your conflict of interest statement in the “Confidential to Editor” section, and submit your "Accept" recommendation.

Reviewer #2: All comments have been addressed

2. Is the manuscript technically sound, and do the data support the conclusions?

Reviewer #2: Yes

3. Has the statistical analysis been performed appropriately and rigorously? 

Reviewer #2: Yes

4. Have the authors made all data underlying the findings in their manuscript fully available?

Reviewer #2: Yes

5. Is the manuscript presented in an intelligible fashion and written in standard English?

Reviewer #2: Yes

6. Review Comments to the Author

Reviewer #2: (No Response)

7. PLOS authors have the option to publish the peer review history of their article (what does this mean?). If published, this will include your full peer review and any attached files.

Reviewer #2: No

---

## [Editor Report · Acceptance letter]

14 Dec 2022

PONE-D-21-33144R2 

Psychopathic tendencies are selectively associated with reduced emotional awareness in the context of early adversity 

Dear Dr. Smith:

I'm pleased to inform you that your manuscript has been deemed suitable for publication in PLOS ONE. Congratulations! Your manuscript is now with our production department. 

Kind regards, 

on behalf of

Dr. Peter Karl Jonason 

Academic Editor

PLOS ONE